# Synthetic gene circuits that selectively target RAS-driven cancers

Gabriel Valentin Senn, Leon Nissen, Yaakov Benenson*†

Department of Biosystems Science and Engineering, ETH Zurich, Basel, Switzerland

## eLife Assessment

This **important** study demonstrates the potential of synthetic gene circuits to detect and target aberrant RAS activity in cancer cell lines. The circuit design is novel and the evidence supporting the claims is **convincing**. As a proof-of-concept, this will be of broad interest to researchers in synthetic biology and therapeutics development, while future work will be required to help translate this technology toward clinical applications in cancer therapeutics and address potential limitations of the strategy.

## Abstract

Therapies targeting mutated rat sarcoma (RAS), the most frequently mutated oncogene in human cancers, could benefit millions of patients. Recently approved RAS inhibitors represent a breakthrough but are limited to a specific KRAS$^{G12C}$ mutation and prone to resistance. Synthetic gene circuits offer a promising alternative by sensing and integrating cancer-specific biomolecular inputs, including mutated RAS, to selectively express therapeutic proteins in cancer cells. A key challenge for these circuits is achieving high cancer selectivity to prevent toxicity in healthy cells. To address this challenge, we present a novel approach combining multiple RAS sensors into RAS-targeting gene circuits, which allowed us to express an output protein in cells with mutated RAS with unprecedented selectivity. We implemented a modular design strategy and modeled the impact of individual circuit components on output expression. This enabled cell-line-specific adaptation of the circuits to optimize selectivity and fine-tune expression. We further demonstrate the targeting capabilities of the circuits by employing them in different RAS-driven cancer cells and provide evidence for their therapeutic potential by linking them to the expression of a clinically relevant output protein, which induced robust killing of cancer cells with mutated RAS. This work highlights the potential of synthetic gene circuits as a novel therapeutic strategy for RAS-driven cancers, advancing the application of synthetic biology in oncology.

**\*For correspondence:**
kobi.benenson@gmail.com

**Present address:** †Pattern Biosciences, Basel, Switzerland

## Introduction

RAS (rat sarcoma) mutations are the most common oncogenic alterations in human cancers (*Timar and Kashofer, 2020*), accounting for 19% of all cases (*Prior et al., 2020*). The RAS gene family consists of HRAS, KRAS, and NRAS, with KRAS being the most frequently mutated isoform (*Muñoz-Maldonado et al., 2019*). Considered 'undruggable' for many years, recent approval of several KRAS$^{G12C}$ inhibitors (*Boumelha et al., 2023*) has demonstrated the efficacy of targeting RAS in cancer treatment. However, these therapies are limited to cancers with KRAS$^{G12C}$ mutations, which narrows the target range and makes the inhibitors susceptible to resistance development due to the emergence of other RAS mutations (*Steffen et al., 2023*). While KRAS$^{G12C}$ remains the only RAS mutation with approved targeted inhibitors, recent advances have expanded the therapeutic landscape. Several inhibitors targeting other RAS mutations, including KRAS$^{G12D}$ (e.g. MRTX1133; *Wei et al., 2024*), as well as

**eLife digest** All cancers are caused by changes in genes. One of the most frequently mutated genes is RAS, which plays a central role in cell signalling. RAS mutations are found in around 19 per cent of cancer patients worldwide and are responsible for nearly 1.9 million cancer-related deaths per year.

Mutated RAS proteins continuously send growth signals to cells, which can cause them to grow uncontrollably and become cancerous. Although recently developed treatments known as RAS inhibitors can target specific RAS cancers, their therapeutic success remains limited.

Synthetic biology – an interdisciplinary field that applies engineering principles to biology – offers a powerful alternative. It enables the design of synthetic gene circuits that can sense and respond to genetic mutations. These circuits can be engineered to detect cancer-specific biomarkers and produce therapeutic proteins that selectively target cancer cells. However, achieving high selectivity for cancer cells while avoiding toxicity in healthy cells remains a major challenge.

To address this problem, Senn, Nissen and Benenson used synthetic biology to develop DNA-encoded therapeutics for RAS-driven cancers. The researchers created two circuits for sensing RAS-specific inputs: sensors that bind directly to RAS and sensors that detect RAS-specific signals. By combining these sensors, they created circuits with unprecedented selectivity for cells carrying RAS mutations, while showing minimal activity in healthy cells.

Moreover, their modular design could be adapted to specific cell lines to optimise selectivity and fine-tune expression levels. The circuits were tested in multiple RAS-driven cancer cell lines and showed therapeutic potential by driving expression of a clinically relevant protein that successfully killed cancer cells with RAS mutations.

Overall, the study by Senn et al. highlights the potential of synthetic gene circuits as a modular and adaptable strategy for targeting cancer. By developing RAS-specific sensors and integrating them into selective circuits, the researchers lay the groundwork for new therapies against RAS-driven cancers that could benefit millions of patients. While clinical application will require improved delivery systems and validation in patient-derived models, this work demonstrates how synthetic gene circuits could help overcome key challenges in cancer therapy.

pan-RAS inhibitors (*Jiang et al., 2024*), are currently in clinical development. A comprehensive overview of these efforts is provided by *Oya et al., 2024*.

Additional RAS-targeting strategies have encompassed protein engineering, RAS silencing, and artificial promoters. Synthetic proteins and endogenous RAS effectors (proteins that interact with active RAS and transduce its signaling) were engineered to bind (*Kiyokawa et al., 2011*), inhibit (*Tomazini and Shifman, 2023*), or degrade RAS (*Bery et al., 2020*; *Lim et al., 2021*). Additionally, miRNAs modulated by oncogenic RAS were identified as potential drug targets (*Shi et al., 2018*) and siRNA against KRAS was used to target RAS-driven cancers in mice (*Pecot et al., 2014*). Artificial promoters or synthetic transcription factor response elements responsive to transcription factors downstream of RAS were developed to interrogate RAS signaling (*Murai and Treisman, 2002*) and target cancer cells with RAS mutations by expressing a toxin-antitoxin system (*Dvory-Sobol et al., 2005*; *Shapira et al., 2021*).

Gene circuits that logically integrate multiple biomolecular inputs are an emergent modality that allows discriminating between malignant and healthy cells and selectively eliciting a therapeutic response only in target cells (*Xie et al., 2011*; *Wu et al., 2019*; *Weber and Fussenegger, 2012*). Logic gene circuit-based therapeutic prototypes already demonstrated efficacy and safety in animal tumor models (*Angelici et al., 2021*). Some of the previously explored approaches for sensing and targeting RAS (*Shi et al., 2018*; *Murai and Treisman, 2002*; *Dvory-Sobol et al., 2005*; *Shapira et al., 2021*; *Lisiansky et al., 2012*) are potentially compatible with the logic gene circuit paradigm. Two circuits have already been shown to build upon, or directly sense, RAS signaling. Gao and colleagues demonstrated a split protease approach to sense over-activation of proteins upstream of RAS. By fusing one domain of a protease to HRAS and the other domain to a RAS-binding domain (RBD) of rapidly accelerated fibrosarcoma type C (CRAF), they created a sensor for overexpressed mutated epidermal growth factor receptor (EGFR) or son of sevenless 1 (Sos-1) in HEK293 cells (*Gao et al.,*

2018). Vlahos and colleagues adapted this approach to directly sense RAS activation by fusing either part of a split protease to an RBD domain, enabling RAS-dependent interleukin-12 secretion in HEK293T cells overexpressing KRASG12V (*Vlahos et al., 2022*). Further advancing synthetic circuits for therapeutic applications against RAS-driven cancer requires achieving high selectivity for mutated RAS and robust function in heterogeneous cancer cells. High selectivity is required to achieve strong output expression in cancer cells while minimizing off-target effects and toxicity in healthy cells with wild-type RAS. In addition, eventual translatability of these approaches to the clinic requires improved robustness in the face of tumor heterogeneity.

In this study, we address these challenges by developing a set of versatile RAS sensors and RAS-dependent transcription factor response elements to create synthetic gene circuits that use RAS activation as input. By combining these direct and indirect RAS sensors in an AND-gate configuration, we develop RAS-targeting circuits with high dynamic range and high selectivity towards cells with mutant RAS. The modular design allows fine-tuning the circuits to specific target and off-target cells and balancing their activation strength versus leakiness. Finally, we demonstrate that our new RAS-targeting circuits function as cancer cell classifiers, selectively expressing an output protein in a wide range of cancer cell lines with RAS-overactivating mutations while maintaining minimal output in cell lines with wild-type RAS.

## Results

### Design of a synthetic RAS sensor

The endogenous RAS signaling pathway is initiated by the binding of cytoplasmic effector proteins to activated RAS-GTP. The effectors sense RAS-GTP dimerization and propagate the signal downstream through a phosphorylation cascade. Post-activation, GTPase-activating proteins, such as neurofibromin 1 (NF-1), facilitate the hydrolysis of RAS-bound GTP to GDP, thereby terminating the signaling. Cancer-associated mutations in RAS render it insensitive to this hydrolysis, resulting in constitutively active RAS (*McFall et al., 2020*) that drives uncontrolled cell proliferation and tumor growth.

Inspired by the natural function of CRAF, a key effector of RAS (*Lavoie and Therrien, 2015*), we designed a sensor that exploits the selective binding to RAS-GTP of CRAF's RBDCRD (**R**AS-**b**inding **d**omain/**c**ysteine **r**ich **d**omain) domain. In our sensor, we fuse the RBDCRD domain to engineered truncated and mutated NarX variants originally derived from the bacterial two-component system, namely NarX$^{379-598}$H399Q and NarX$^{379-598}$N509A (*Figure 1a*). We showed previously that these NarX variants are able to transphosphorylate in mammalian cells, but only upon forced dimerization via fused protein domains. Therefore, NarX variants can functionally replace CRAF's own dimerization domain when fused to RBDCRD, while enabling orthogonal signaling in mammalian cells via a humanized NarL response regulator (*Mazé and Benenson, 2020*). We call these chimeric constructs 'RBDCRD-NarX fusions'.

We surmised that the RAS sensor would function as follows: activation of RAS (whether endogenous or mutation-driven) would elevate RAS-GTP levels, which would in turn bind the RBDCRD domain of the RBDCRD-NarX fusion. Resulting dimerization of RBDCRD would lead to a forced dimerization of the fused NarX, leading to NarX$^{H399Q}$ transphosphorylating NarX$^{N509A}$, in turn phosphorylating NarL. Phosphorylated NarL would bind to its response element on the NarL-responsive promoter and induce the expression of an output protein, here mCerulean (*Figure 1b*). In order to test these assumptions, we constructed the necessary components including the two complementary RBDCRD-NarX fusions, the humanized NarL and the NarL-responsive promoter coupled to mCerulean. Initially, we chose HEK293 cells as a test system. To emulate the presence of mutant RAS, we transfected HEK293 cells with a plasmid expressing KRAS$^{G12D}$, and to emulate high levels of wild-type RAS, we transfected them with a plasmid-encoded wild-type KRAS (KRAS$^{WT}$). (Note that HEK293 cells express low endogenous levels of wild-type KRAS, HRAS, and NRAS.) In the first set of tests, delivery of all the sensor components to HEK293 cells using transient transfection showed significantly higher output expression in cells expressing KRAS$^{G12D}$ than in cells transfected with KRAS$^{WT}$ (*Figure 1c*). In HEK293 cells transformed with the mutant KRAS, sensor response increased upon increase in the dose of the sensor-encoding plasmids (*Figure 1d*). Further, sensor function depended on RAS binding, because mutations in the RBD and the cysteine-rich domain (CRD) strongly decreased output expression. To demonstrate this, we mutated two residues in RBDCRD important for RAS-RAF signaling:

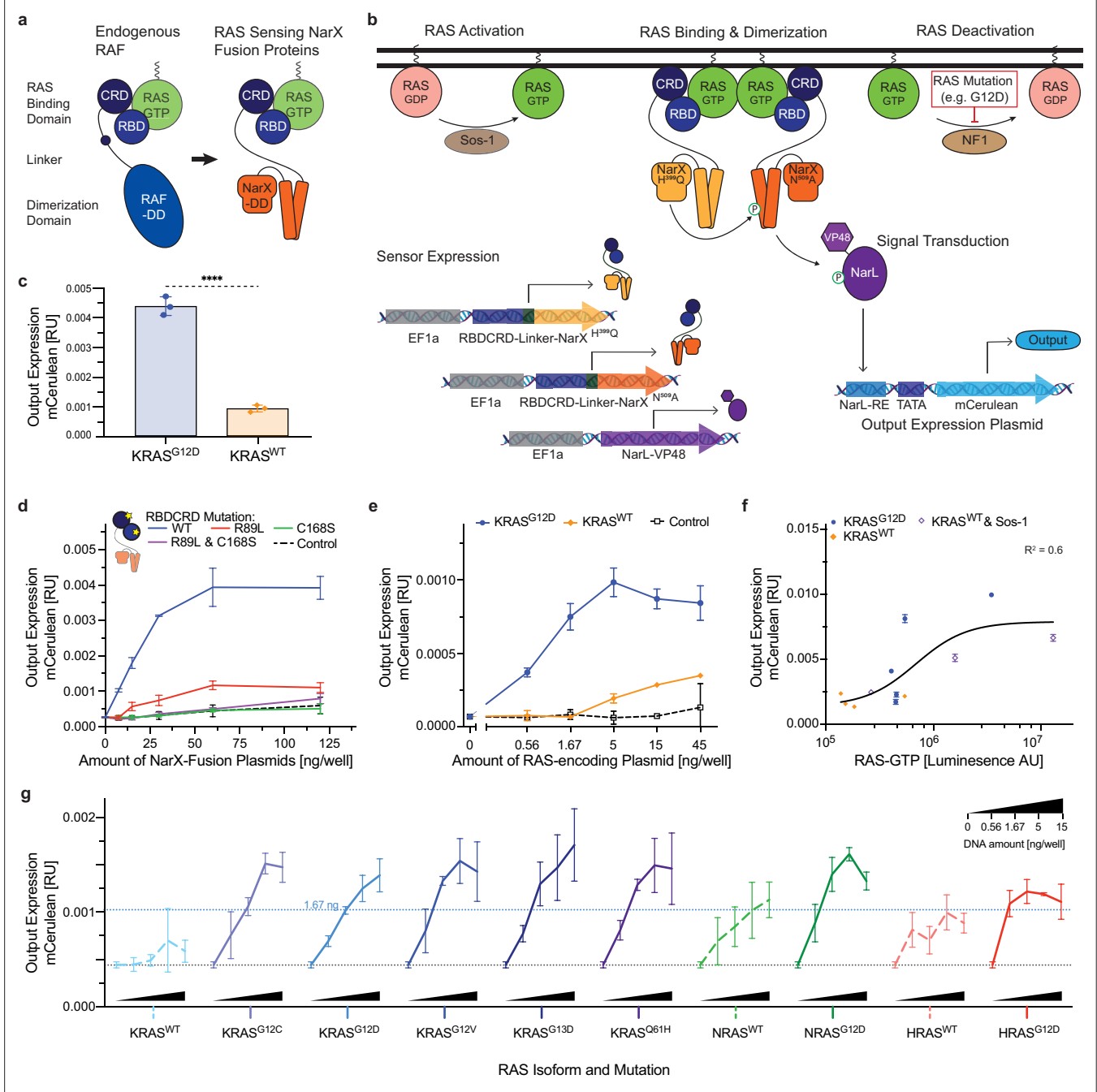

**Figure 1.** Design and characterization of the RAS sensor. (**a**) Design of the binding component of the RAS sensor. Inspired by natural RAF (left), the RAS-binding component of the sensor (right) comprises RAS-binding domain (RBDCRD), a linker, and a NarX-derived transphosphorylation domain. (**b**) Schematic of the RAS sensor composition and mechanism of action. The sensor's genetic payload is encoded on four plasmids. Two plasmids express the RAS-binding components: RBDCRD fused, respectively, to NarXN509A or NarXH399Q. The third plasmid expresses NarL-VP48, and the fourth plasmid encodes the output protein (mCerulean) under the control of a NarL response element (NarL-RE) in front of a minimal promoter (TATA). Upon RAS activation, the RBDCRD domain of the RAS-binding components binds to RAS-GTP. This binding leads to a forced dimerization of the NarX domains and a transphosphorylation of NarXN509A, in turn phosphorylating NarL. Phosphorylated NarL binds its response element on the output plasmid, inducing the expression of the output protein. (**c**) Sensor activation by mutated RAS. The bar chart shows output expression in HEK293 cells co-transfected with the RAS Sensor and either KRAS$^{G12D}$ or KRAS$^{WT}$. (**d**) Dose-response curve and dependence of the RAS sensor on functional RAS binding. Output expression of RAS sensors with either RBDCRD wild-type (blue) or RBDCRD with R89L (red), C168S (green), or both (purple) mutations. The dashed line represents conditions where the NarX-fusion plasmids were replaced with a non-coding plasmid (control). (**e**) Dependence of sensor output on RAS levels. Output expression of the RAS sensor measured with increasing amounts of KRAS$^{G12D}$ (blue), KRAS$^{WT}$ (orange), or negative control (black) plasmid. (**f**) Input-output curve. Correlation of the output expression with the RAS-GTP levels in HEK293 cells, measured by a luminescence

*Figure 1 continued on next page*

*Figure 1 continued*

RAS-pulldown ELISA assay. To alter RAS-GTP levels, the cells were transfected with different amounts of either KRAS$^{G12D}$ (blue), KRAS$^{WT}$ (orange), or KRASW$^T$ + Sos-1 (purple) plasmids. Pearson's correlation is shown as R$^2$. (**g**) Generalizability across RAS variants. Output expression of the RAS sensor when co-transfecting increasing amounts of different RAS isoforms and mutants. mCerulean output expression was measured by flow cytometry and normalized to a constitutively expressed mCherry transfection control. Mean values were calculated from biological triplicates. Error bars represent +/- SD. Significance was tested using an unpaired two-tailed Student's t-test. ****p<0.0001.

The online version of this article includes the following figure supplement(s) for figure 1:

**Figure supplement 1.** RAS sensor activation with RBD or RBDCRD as binding domain.

**Figure supplement 2.** RAS isoforms and mutants.

Arginine 89 in RBD (*Fabian et al., 1994*) and Cysteine 168 in CRD (*Roy et al., 1997*). Arginine 89 forms electrostatic interactions with acidic residues in the RAS switch I region, stabilizing the RBD–RAS complex. Cysteine 168 is part of a zinc finger motif that is critical for high-affinity association with RAS and efficient RAF signaling (*Tran et al., 2021*; *Luo et al., 1997*). Mutating these residues –R89L in RBD or C168S in CRD– has been shown to reduce RAS binding, diminish RAF kinase activity (*Tran et al., 2021*; *Luo et al., 1997*), and impair RAS-dependent membrane localization of RBD–CRD fusion proteins (*Bondeva et al., 2002*). Interestingly, while the RBD$^{R89L}$ mutant still exhibited dose-dependent leakiness, the CRD$^{C168S}$ and the double RBD$^{R89L}$CRD$^{C168S}$ mutant reduced output expression to background levels (*Figure 1d*). In agreement with previous reports (*Tran et al., 2021*) and with our observation that the sensor employing RBDCRD instead of RBD alone shows increased downstream signaling (*Figure 1—figure supplement 1*), this result confirms that both RBD and CRD domains are involved in RAS binding and sensor activation.

We then characterized sensor response to increasing input levels. As expected, increased levels of mutant KRAS drove higher output expression, whereas wild-type KRAS resulted in some sensor activation but at much higher expression levels (*Figure 1e*). Finally, we measured the RAS-GTP protein levels in HEK293 cells using a RAS-pulldown ELISA assay. We manipulated the RAS-GTP levels in the cells by co-expressing different amounts of KRAS$^{WT}$, KRAS$^{G12D}$, or KRAS$^{WT}$ +Sos-1, a guanine nucleotide exchange factor that activates RAS, and were able to directly correlate higher RAS-GTP levels with higher output expression (*Figure 1f*).

To assess the generalizability of the RAS sensor across different oncogenic mutations and RAS isoforms, we tested a panel of oncogenic RAS variants, including multiple KRAS mutants as well as HRAS$^{G12D}$ and NRAS$^{G12D}$. In all cases, we observed concentration-dependent activation of the sensor (*Figure 1g*). Among the KRAS mutants, no significant differences in sensor output were observed (*Figure 1—figure supplement 2a–b*), suggesting that the behavior characterized with KRAS$^{G12D}$ is representative of other activating KRAS mutations. In contrast, we found more heterogeneous responses across RAS isoforms (*Figure 1—figure supplement 2c–f*). Notably, high overexpression of wild-type HRAS or NRAS resulted in stronger sensor activation than wild-type KRAS (*Figure 1—figure supplement 2f*). This indicates that all wild-type RAS isoforms can activate the sensor, underscoring the need to consider physiological activity of all isoforms when evaluating circuit specificity and minimizing potential off-target effects in healthy cells.

Collectively, this dataset provides evidence that the RAS sensor is activated by RAS-GTP and therefore can selectively sense mutant RAS variants because the hallmark of these mutants is increased RAS-GTP levels. Nonetheless, wild-type RAS also binds GTP, which explains sensor response in HEK293 cells that overexpress wild-type RAS and contain relatively high levels of RAS-GTP (*Figure 1f*).

## Mechanism of action

To recapitulate, our design anticipates that the function of the RAS sensor requires the following steps: delivery and expression of the RAS sensor components; RAS-GTP binding of RBDCRD-NarX fusion proteins via their RBDCRD domain, resulting in forced dimerization of the NarX domains and transphosphorylation; NarL phosphorylation; and NarL-mediated output expression. To quantify the effect of RAS activation on these steps, we manipulated the RAS-GTP levels in HEK293 cells in a variety of ways: (i) we overexpressed wild-type KRAS to show the effect of high non-mutant KRAS concentration on downstream sensor response and quantify sensor activation in the presence of RAS-GTP associated with wild-type KRAS; (ii) we overexpressed mutant KRAS in order to measure sensor response to the mutant KRAS input; (iii) we overexpressed NF1, an endogenous GTPase-activating

protein that deactivates residual endogenous wild-type RAS in HEK293 cells and allows us to quantify non-specific sensor response in the absence of RAS-GTP; and (iv) we expressed Sos-1, which activates endogenous RAS generating high levels of RAS-GTP, to quantify the upper bound on sensor output without RAS overexpression.

In order to quantify signal propagation along this cascade, we generated a number of reporter constructs. First, to quantify the expression of the RBDCRD-NarX fusion, we further fused this component to SYFP2 and measured YFP fluorescence in HEK293 cells with various RAS-GTP levels. We observed that overexpression of KRAS$^{G12D}$ or overactivation of endogenous RAS with Sos-1 resulted in YFP increase and thus elevated expression of the RBDCRD-NarX fusion, compared to cells with baseline RAS-GTP. Conversely, inactivation of endogenous RAS with NF1 resulted in a lower YFP/RBDCRD-NarX expression (**Figure 2a**). Second, we used the same construct to visualize RAS binding of the RBDCRD-NarX. Because RAS is a membrane protein (**Nussinov et al., 2020**), we expected the binding of RBDCRD-NarX to KRAS-GTP to result in YFP accumulation at the membrane. Indeed, RAS-GTP increase by overexpressing KRAS$^{G12D}$ or via Sos-1 led to higher membrane-to-total-signal ratio, compared to cells with endogenous RAS-GTP levels or with NF1-deactivated RAS (**Figure 2b**). Third, we examined the dimerization of the RAS sensor by fusing the two parts of a split mVenus to RBDCRD-NarX proteins and measuring the signal from reconstituted mVenus. Again, RAS activation increased the signal from dimerized RBDCRD-NarX-mVenus, while RAS deactivation decreased it (**Figure 2c**). Last, we investigated the effect of RAS-GTP levels on mCerulean sensor output of the complete RAS sensor. We observed the expected trend, whereas the magnitude of the effect was amplified compared to the effects of individual sensing step, suggesting signal amplification in the synthetic sensing pathway (**Figure 2d**).

While the dependency of RAS binding, dimerization, and output expression of the sensor on RAS-GTP levels was expected, we did not expect that the expression of the RBDCRD-NarX fusion itself (**Figure 2a**) would be RAS-GTP dependent. Possible explanations could include the presence of transcription factor binding sites activated by RAS signaling in the elongation factor 1 a (EF1a) promoter driving the expression of RBDCRD-NarX, or RAS-dependent non-specific activation of protein synthesis. EF1a promoter sequence contains potential binding sites of CREB, c-Myc, SRF, AP1, and Elk-4 (**Figure 2—figure supplement 1** and **Figure 2—source data 1**; **Lee and Huang, 2014**). Additionally, there are reports that (over-)activated RAS or MAPK signaling can increase protein synthesis (**Proud, 2004**; **Rinker-Schaeffer et al., 1992**; **Lavoie et al., 2020**; **Wang and Proud, 2002**) through post-transcriptional and translational mechanisms. These include upregulation of translation initiation factors and translational machinery (**Proud, 2004**), as well as increased ribosome biogenesis (**Proud, 2004**; **Azman et al., 2023**), suggesting that multiple mechanisms may be involved.

Comparing the functional RAS sensor to a panel of non-functional or constitutively active control sensors resulted in two additional insights: One, inactive controls lacking NarX dimerization show low increases in mCerulean output (**Figure 2—figure supplement 2a**), which are fully compensated – or slightly overcompensated – by normalization to the EF1a-driven mCherry transfection control (**Figure 2—figure supplement 2b**). Two, the RAS-dependent increase in expression is non-linearly amplified by the NarX-NarL system. Although the resulting fourfold RAS-dependent increase in output signal is significantly lower than the 14-fold increase of the functional RAS sensor, suggesting that increased expression does not explain the full response (**Figure 2—figure supplement 2b**).

This indicates that dimerization and functional binding of RBDCRD to RAS are required for full sensor activation and output generation, which is further evidenced by the decreased activity of non-RAS binding mutants (**Figure 1e** and **Figure 2—figure supplement 2**). While increased expression and its non-linear amplification are a contributing factor, RAS binding and RAS-dependent dimerization are necessary for achieving maximal dynamic range in RAS sensing.

## Optimization of activation efficiency and sensor tunability

The RAS sensor is activated by RAS-GTP and therefore by both wild-type and mutant RAS. Cells with mutant RAS have higher RAS-GTP and higher sensor activation than cells with wild-type RAS. Nevertheless, residual sensor activation in healthy cells with wild-type RAS may cause toxicity in a clinical implementation of the sensor. Therefore, we aimed to further improve the RAS sensor's ability to discriminate between wild-type and mutant RAS and optimized the transfer function between RAS-GTP input and mCerulean output. Structural predictions with AlphaFold2 (**Jumper et al., 2021**)

motivated the exploration of longer linkers between the sensor's RBDCRD and NarX-derived domains by showing that longer linkers may provide more flexibility for the NarX domains to get into close proximity, which could improve dimerization and transphosphorylation efficiency (*Figure 3a*). Our measurements showed that up to a certain size, longer and/or more rigid linkers indeed increased output expression, but not the dynamic range between KRAS$^{G12D}$ and KRAS$^{WT}$ (*Figure 3b*).

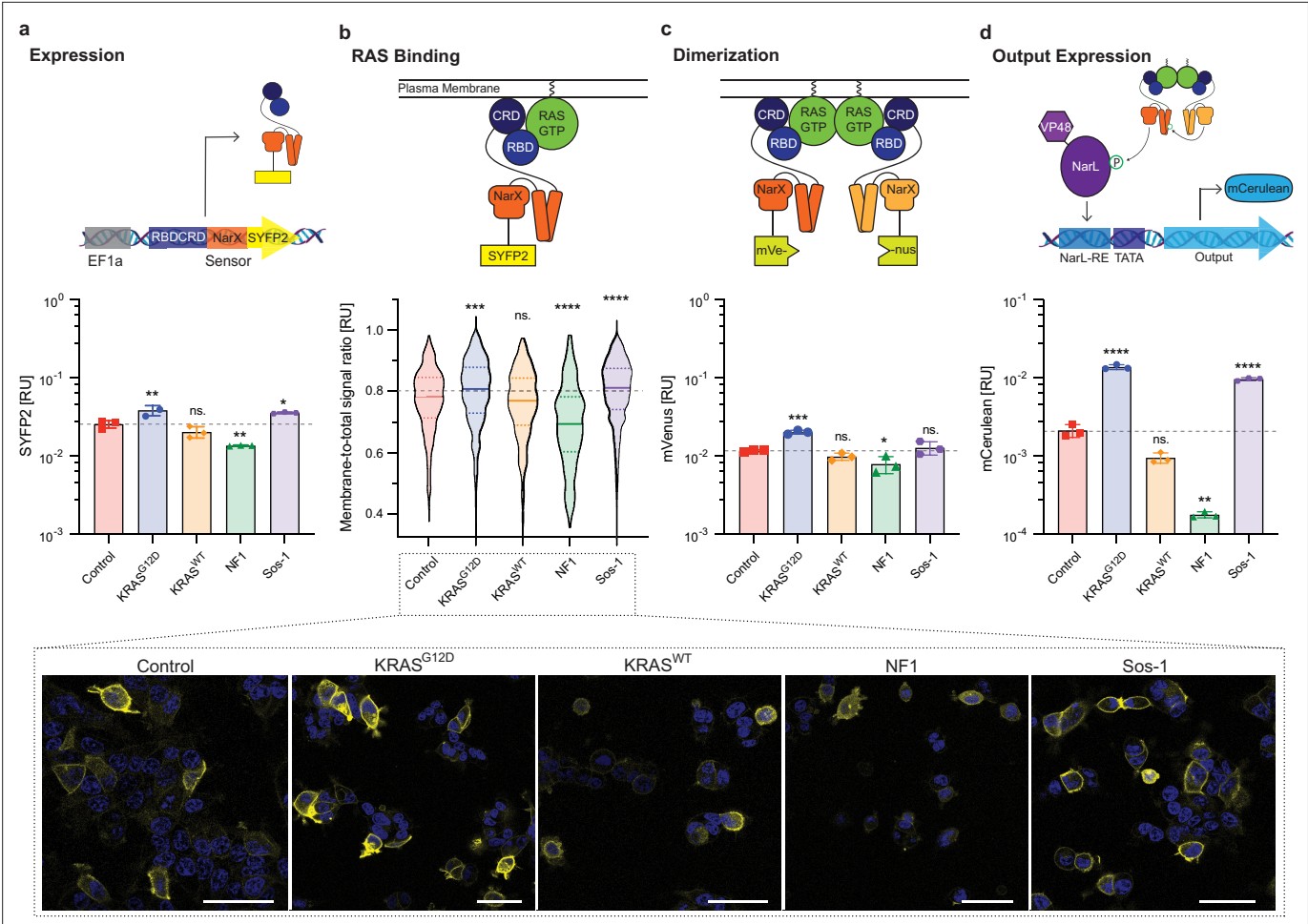

**Figure 2.** Mechanism of action. Effect of differential RAS activation on the steps considered necessary for RAS Sensor activation. RAS activation in HEK293 cells was manipulated by co-expressing KRAS$^{G12D}$, KRAS$^{WT}$, Sos-1 (a guanine nucleotide exchange factor that activates endogenous RAS), or NF1 (a GTPase-activating protein that deactivates endogenous RAS). In the control condition, the cells are transfected with a non-coding plasmid; here, it represents the endogenous RAS activation. Schematics on top of the graphs illustrate how and what part of the Mechanism was investigated. (**a**) Expression levels of the RBDCRD-NarX-SYFP2 fusion protein measured by flow cytometry in the presence of various KRAS modulators (x axis labels). (**b**) RAS binding of the RBDCRD-NarX-SYFP2 fusion protein approximated by calculating the ratio of membrane to total SYFP2 signal for each cell. Intracellular localization of SYFP2 was measured using confocal microscopy. The micrographs below show representative images for each condition. Scale bars = 50 μm. (**c**) Dimerization of the NarX fusion proteins assessed by transfecting two complementary RBDCRD-NarX-split.mVenus fusions and measuring the mVenus fluorescence by flow cytometry. (**d**) Output expression after transfection with the full RAS Sensor measured by flow cytometry. In **a**, **c**, and **d**, the fluorescent signals were normalized to a constitutively expressed transfection control. Each symbol represents one biological replicate (**a**: n=9, **c-d**: n=3). The error bars represent +/- SD. In **b**, the fluorescence at the membrane was normalized to the total fluorescence for each cell. Violin plots in **b** represent 560 (KRAS$^{G12D}$), 322 (KRAS$^{WT}$), 482 (Control), 226 (NF1), or 1194 (Sos-1) cells from three biological replicates. Significance was tested using an ordinary one-way ANOVA with Dunnett's multiple comparisons to compare each condition with the control condition (endogenous RAS activation). *p<0.05, **p<0.01, ***p<0.001, ****p<0.0001.

The online version of this article includes the following source data and figure supplement(s) for figure 2:

**Source data 1.** Prediction of MAPK transcription factor binding sites in the EF1a promoter.

**Figure supplement 1.** Prediction of MAPK transcription factor binding sites in EF1a.

**Figure supplement 2.** Contribution of RAS-dependent increase in sensor expression levels on total output.

We also explored alternative natural and synthetic RBDs. Among the tested binding domains, the Ras association domain (RA) of the natural RAS effector Rassf5, the RAS association domain 2 (RA2) of the phospholipase C epsilon (PLCe; *Nakhaeizadeh et al., 2016*), and the synthetic RAS binder K55 (*Guillard et al., 2017*) showed a slightly higher or similar dynamic range than RBDCRD (*Figure 3c*). To better understand how different binding domains affect RAS-specific activation, we deactivated endogenous RAS$^{WT}$ in HEK293 using NF-1. RBDCRD was the only binding domain that showed a pronounced decrease in output expression in cells with NF1 compared to cells with KRAS$^{WT}$, indicating low RAS-unspecific activation for RBDCRD but also activation in cells with KRAS$^{WT}$. The other binding domains showed one of two responses: (1) already very low activation in cells with KRAS$^{WT}$ which was

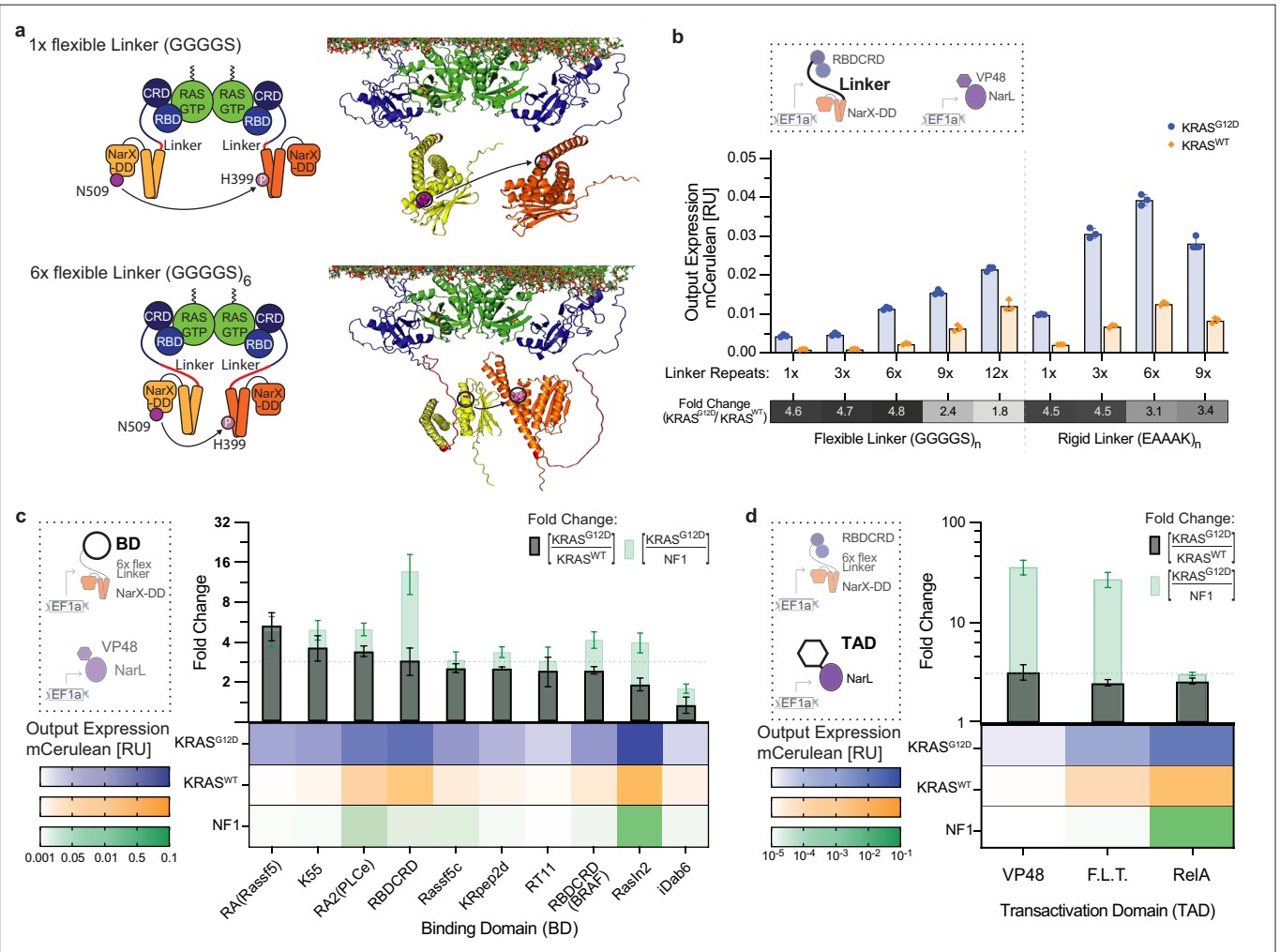

**Figure 3.** Tunability of the RAS sensor. (**a**) 3D structure of the RAS sensor dimerizing at the membrane. The structure of the RBDCRD-NarX fusion proteins (orange and yellow) was predicted using AlphaFold and aligned with existing NMR structures of RBDCRD (blue) bound to a KRAS-dimer (green) at the membrane. The ATP-binding site N509 (purple) and the phosphorylation site H399 (pink) are highlighted as spheres. On the top, RBDCRD and NarX are fused with a 1 x GGGGS linker and on the bottom with a 6 x GGGGS linker. (**b**) Effect of different linkers in the RBDCRD-NarX fusion protein of the RAS Sensor. The bar charts show the output expression in HEK293 co-transfected with KRAS$^{G12D}$ (blue) or KRAS$^{WT}$ (orange) when using different numbers of repeats of a flexible (GGGGS) or a rigid (EAAAK) linker in the RBDCRD-NarX fusion proteins. The heatmap below shows the corresponding fold change between output expression in cells with KRAS$^{G12D}$ and KRAS$^{WT}$. (**c**) Effect of different binding domains (BD) fused to NarX in the RAS Sensor. The heatmap shows the output expression in HEK293 co-transfected with 15 ng/well of KRAS$^{G12D}$ (blue), KRAS$^{WT}$ (orange), or NF1 (green), a GTPase-activating protein that deactivates endogenous RAS. The bars above show the corresponding fold changes between cells with KRAS$^{G12D}$ and KRAS$^{WT}$ (black) or KRAS$^{G12D}$ and NF1 (green). (**d**) Effect of different transactivation domains (TAD) fused to NarL in the RAS Sensor. The heatmap shows the output expression in HEK293 co-transfected with KRAS$^{G12D}$ (blue), KRAS$^{WT}$ (yellow), or NF1 (green). The bars above show the corresponding fold changes between cells with KRAS$^{G12D}$ and KRAS$^{WT}$ (black) or KRAS$^{G12D}$ and NF1 (green). mCerulean output expression was measured using flow cytometry and normalized to a constitutively expressed mCherry transfection control. Mean values were calculated from three (**b**) or two (**c–d**) biological replicates. Error bars were calculated using error propagation rules.

not further decreased in cells with NF1, for example in RA(Rassf5) and K55; (2) background activation in cells with NF-1 indicating RAS-unspecific activation, for example in RA2(PLCe) (*Figure 3c*).

Fusing stronger transactivation domains to NarL markedly increased output expression of the RAS sensor without changing its dynamic range (*Figure 3d*). Specifically, we replaced the initial VP48 with either F.L.T., a fusion of three partial transactivation domains, FoxoTAD (Forkhead-Box-Protein-O3$^{604-664}$), LMSTEN (MYB$^{251-330}$), and TA1 (RelA$^{521-331}$) or with RelA$^{342-551}$ with all its transactivation domains. In cells co-transfected with NF1, the sensor with NarL-F.L.T. exhibited low output expression similar to the initial sensor variant with NarL-VP48. In contrast, the sensor with NarL-RelA had high background in cells with NF1, suggesting RAS-unspecific activation when using RelA as a transactivation domain (*Figure 3d*). Overall, the testing of alternative sensor parts did not increase the dynamic range between KRAS$^{G12D}$ and KRAS$^{WT}$, but it showed that the RAS sensor was modular. Different linkers, binding domains, and transactivation domains could be used to construct the sensor and tune the absolute strength of output expression without affecting the dynamic range.

## Multi-input RAS-targeting circuits with improved selectivity for KRAS$^{G12D}$

Mutations in signaling proteins, such as RAS, often lead to increased activation of downstream transcription factors (*Saliani et al., 2019*). We hypothesized that using transcription factors from the mitogen-activated protein kinase pathway (MAPK) downstream of RAS as inputs could add an additional layer of selectivity to RAS-targeting circuits. Regulating the expression of RAS sensor components via MAPK-dependent response elements allowed us to create a coherent type-1 feed-forward motif with AND-gate logic (*Alon, 2007*; *Figure 4a*). This motif has been shown to act as a noise repressor (*Pieters et al., 2021*) and as a persistence detector with a delayed onset that is only activated by persisting input stimulation (*Alon, 2007*). Both are desired properties in RAS-targeting circuits to detect constitutively active mutated RAS while minimizing output expression from the more transient activation of wild-type RAS (*Coyle and Lim, 2016*; *Li et al., 2012*).

As an initial step, we designed synthetic response elements with binding sites for transcription factors shown to be activated by mutated RAS, including Elk-1 (*Shi et al., 2018*), c-Fos & c-Jun (*Mechta et al., 1997*), c-Myc (*Vaseva et al., 2018*), and SRF (*Yi et al., 2018*). For each response element, multiple binding sites of MAPK transcription factors (*Murai and Treisman, 2002*; *Kato et al., 1992*; *Gazon et al., 2018*), minimal promoters from genes downstream of RAS (*Murai and Treisman, 2002*), or sequences from existing RAS-responsive promoters (*Dvory-Sobol et al., 2005*; *Lisiansky et al., 2012*; *Nissim et al., 2017*) were encoded upstream of a low leakage minimal promoter (*Hansen et al., 2014*; *Figure 4b* and detailed in *Figure 4—figure supplement 1*).

To assess the functionality of the response elements, we cloned the response elements directly upstream of a fluorescent mScarlet protein and transfected them into HEK293 cells. The minimal promoter of c-fos (pFos; *Murai and Treisman, 2002*), the polyomavirus enhancer domain (PY2; *Lisiansky et al., 2012*), and the serum response element (SRE; *Murai and Treisman, 2002*) were the response elements exhibiting the highest dynamic range between cells with KRAS$^{G12D}$ and cells with KRAS$^{WT}$ (*Figure 4c*). The response elements demonstrated a similar dependency on KRAS$^{G12D}$ and KRAS$^{WT}$ to the one observed with the binding-based RAS sensor in *Figure 1e* and *Figure 4d*. All three response elements contain binding sites for Ets-like 1 protein (elk-1). However, the Elk response element consisting of only elk binding sites showed low mScarlet expression and low dynamic range. To better understand this apparent contradiction, we further investigated the Elk response element. Overexpressing elk-1 with the Elk response reporter increased the mScarlet expression, indicating that endogenous elk-1 levels are not high enough for efficient activation of the Elk response element. With overexpressed elk-1, we observed high selectivity for cells with KRAS$^{G12D}$ and low RAS-independent background activation in cells overexpressing NF-1 (*Figure 4—figure supplement 2*). A RAS titration provided more evidence for the RAS dependence of the Elk response element (*Figure 4e*).

Superior performance of pFos, SRE, and PY2 response elements demonstrates that inclusion of additional transcription factor binding sites can amplify transcriptional sensor response, potentially due to synergy (*Angelici et al., 2016*). Consistent with this and previous reports that the combination of AP-1 and elk-1 binding sites is critical for maximal RAS responsiveness in the polyomavirus enhancer (*Dvory-Sobol et al., 2005*; *Wasylyk et al., 1990*), we found that combining elk-1 with additional transcription factors allows the design of efficient RAS-dependent response elements.

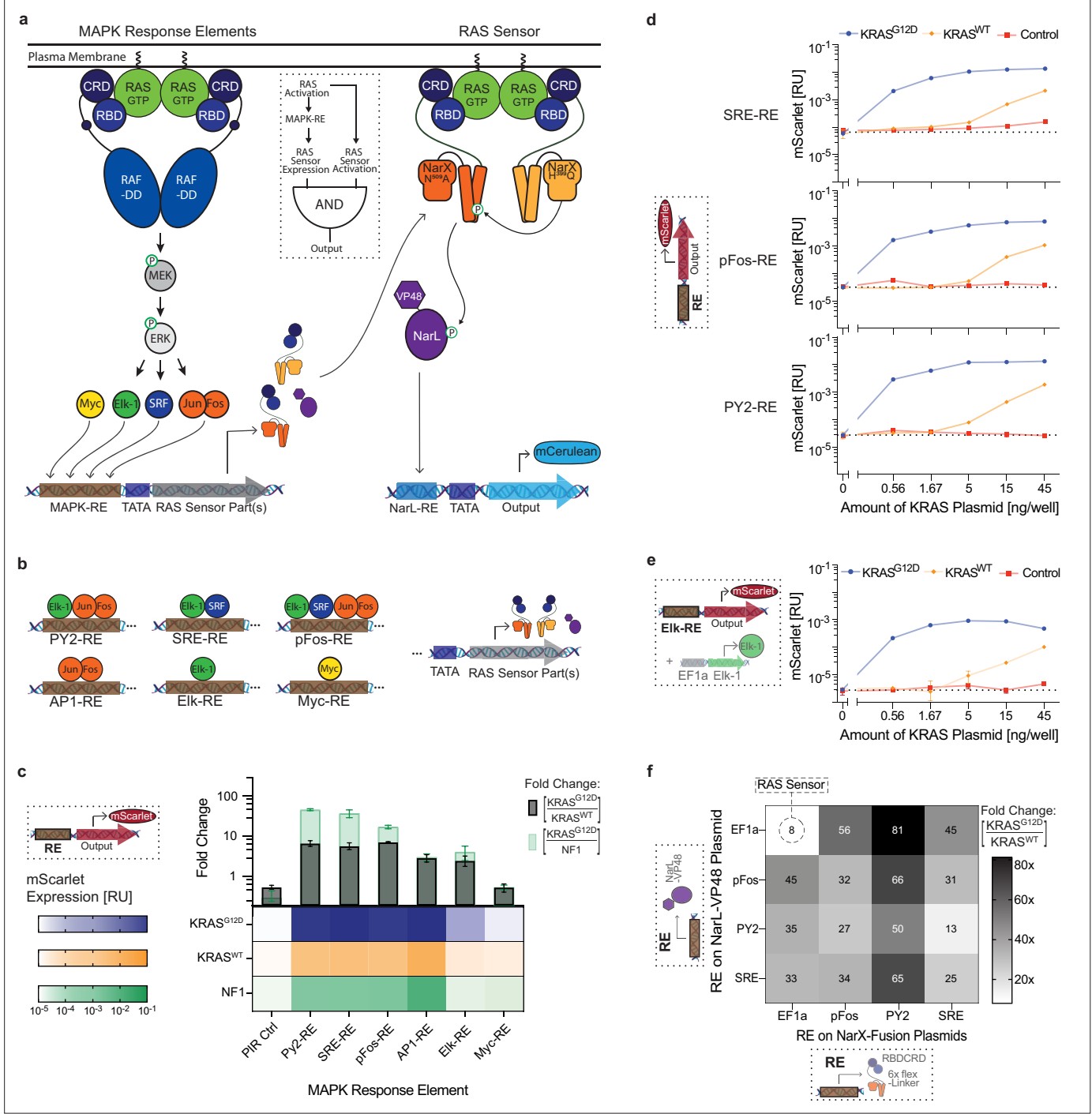

**Figure 4.** Design of multi-input RAS-targeting circuits. (**a**) Schematic of the RAS-targeting circuit with an AND gate between mitogen-activated protein kinase (MAPK) sensors and the direct RAS sensor. Dimerization of RAS activates the MAPK pathway and its downstream transcription factors. These transcription factors bind the synthetic response elements (RE), expressing the parts of the RAS Sensor. The RBDCRD-NarX proteins then bind activated RAS, dimerize, and propagate the signal to NarL, leading to output expression. The logic diagram of the resulting coherent feed-forward loop with AND-gate logic is shown in the dotted box. (**b**) Schematic of the transcription factor binding sites present in the response elements. Multiple repeats of the binding sites were placed upstream of a minimal promoter (TATA) driving expression of the RAS sensor parts. (**c**) Expression levels with the MAPK response elements. The heatmap shows the direct expression of mScarlet of the different REs in HEK293 cells co-transfected with 15 ng/well of either KRAS$^{G12D}$ (blue), KRAS$^{WT}$ (orange), or NF1, a protein that deactivates endogenous RAS (green). The bars above show the corresponding fold changes between cells with KRAS$^{G12D}$ and KRAS$^{WT}$ (black) or KRAS$^{G12D}$ and NF1 (green). (**d**) RAS dependency of the SRE-, pFos-, and PY2- response elements. Direct mScarlet expression of the response elements in HEK293 cells co-transfected with different amounts of KRAS$^{G12D}$ (blue), KRAS$^{WT}$ (orange), or non-

*Figure 4 continued on next page*

Figure 4 continued

coding control plasmids (red). (**e**) RAS-dependency of the Elk-RE when additionally overexpressing Elk-1. RAS titration as described in **d**. (**f**) Dynamic range of the RAS-targeting circuits. Fold change in mCerulean output expression between HEK293 co-transfected with 1.67 ng/well of KRAS[G12D] and KRAS[WT]. In the RAS-targeting circuits, NarL-VP48 and/or the RBDCRD-6xfL-NarX fusion proteins were expressed using different MAPK-REs or a constitutive promoter (EF1a). Fluorescent protein expression was measured by flow cytometry and normalized to a constitutively expressed transfection control. Mean values were calculated from two (**c–e**) or three (**f**) biological replicates. PY2: polyoma virus enhancer domain; SRE: Serum response element; pFos: minimal promoter of c-fos; AP1: activator protein 1; Elk: Ets-like protein; Myc: myelocytomatosis protein. Detailed response element design is shown in *Figure 4—figure supplement 1*.

The online version of this article includes the following figure supplement(s) for figure 4:

**Figure supplement 1.** Design of the MAPK response elements.

**Figure supplement 2.** MAPK response element with overexpressed transcription factors.

**Figure supplement 3.** Comparison of the RAS-binding dependent AND-gate RAS circuits and MAPK response element expressed constitutively dimerized NarX-NarL TCS.

Finally, we combined these MAPK response elements with the RAS sensor using AND-gate logic, by using the response elements to regulate the expression of binding-triggered RBDCRD-NarX sensor. The resulting RAS-sensing circuits showed improved selectivity for KRAS[G12D] with the best performer exhibiting an 81-fold dynamic range between KRAS[G12D] and KRAS[WT] compared to the 8-fold dynamic range of the binding-based RAS sensor alone (*Figure 4f*). To assess the importance of RAS-binding in the RAS-targeting circuits, we replaced the RBDCRD-NarX fusion proteins with a constitutively dimerized non-truncated NarX control. The MAPK response element-expressed, non-truncated NarX showed dynamic ranges between KRAS[G12D] and KRAS[WT] from 2-fold to 20-fold. However, expressing RAS-binding dependent RBDCRD-NarX showed a higher dynamic range for all but the SRE_NarX +PY2_NarL combination (*Figure 4—figure supplement 3*). Thus, combining the MAPK response elements with the binding-based RAS sensor into RAS-targeting circuits generally improved the distinction between cells with KRAS[G12D] and KRAS[WT] and allowed for higher maximal fold changes.

## Modularity allows creating RAS sensor circuits with high dynamic range

As shown above, RAS-sensing circuits can utilize a variety of MAPK response elements, RBDs, linkers, and transactivator domains. To further optimize the transfer function between the input and the output and improve the discrimination between cells with mutant and wild-type RAS, we performed a screening campaign to examine the effect of different combinations of circuit components on the dynamic range. In this screen, we tested combinations of various building blocks: RBDCRD, K55, and RA(Rassf5) as binding domains fused to the NarX; VP48 or F.L.T as transactivation domain of NarL; and pFos, PY2, and SRE as MAPK response elements. The response elements regulated the expression of either the NarX fusion proteins only or of the NarL protein only, while the other part was constitutively driven by EF1a. For the EF1a-driven RBDCRD-NarX proteins, we tested two versions with either a 6 x flexible or a 3 x rigid linker. This set resulted in a total of 222 unique conditions, over two experimental batches. Almost thirty conditions resulted in high dynamic ranges of >100 fold, with the highest circuit variants using the combination of PY2, RBDCRD, and F.L.T. (*Figure 5a*).

We found that PY2 was the most prevalent response element among the high-performing hits, followed by pFos, while SRE only occurred in three hits. The prevalence of the transactivation domain varied depending on whether the NarX or the NarL proteins were regulated by an MAPK response element. While F.L.T. was more abundant among the hits where NarL was regulated by the response elements, VP48 dominated in the circuits where the response elements regulated the NarX fusion proteins. The K55 and RBDCRD binding domains were found in an equal number of hits (*Figure 5b*). However, we observed that circuits with F.L.T. as the transactivation domain performed best with RBDCRD as the binding domain, while there was no significant difference between RBDCRD and K55 for circuits with VP48 (*Figure 5—figure supplement 1*).

To further improve our understanding of the effect of different building blocks, we fitted a linear regression model (*Figure 5c*) and compared it with the experimental data (*Figure 5—figure supplements 2 and 3*). This revealed that the effect of the MAPK response elements depended on which sensor components they regulated. Expression of NarL-F.L.T. under the control of MAPK response elements strongly increased the output expression in cells with KRAS[G12D], while, except for the

SRE_NarL-F.L.T. circuit, the activation in cells with KRAS[WT] remained low. In contrast, MAPK control of NarL-VP48 decreased the output in KRAS[G12D] and even more so in KRAS[WT] cells. Expression of the NarX fusion proteins increased the circuit activation in cells with KRAS[G12D], but also in cells with KRAS[WT]. (*Figure 5c*, & *Figure 5—figure supplement 2a–f*).

In this campaign, we focused on circuits where only NarL or only the NarX proteins were regulated by the response elements. Next, we also implemented the regulation of all circuit components by PY2. The PY2_NarX and NarL-F.L.T. circuit led to the highest output in cells with KRAS[G12D], with only marginally higher activation with KRAS[WT] than PY2_NarX_F.L.T. or PY2_NarL-F.L.T. and still lower than the RAS_Sensor_F.L.T. (*Figure 5d–e*).

In summary, the tests revealed that the RAS-targeting circuits are modular, with various combinations of parts leading to circuits with high dynamic ranges that diverge in their transfer functions and their activation thresholds. The availability of different circuit parts thus allowed for the adaptation of the input-output behavior of the circuits and increased the distinction between cells with mutated and cells with wild-type RAS.

## RAS sensor circuits detect endogenous RAS mutations in cancer cells

To evaluate the response of the RAS sensor circuits to endogenous RAS activation in cancer cell line, we transfected the EF1a-driven, binding-triggered RAS sensor into wild-type HCT-116 cells, a colon cancer cell line harboring a KRAS[G13D] mutation (HCT-116[WT]; *Bairoch, 2018*). To have a comparable off-target cell line without mutated RAS, we also transfected a commercially available KRAS knock-out HCT-116 cell line (HCT-116[k.o.]).

The RAS sensor was functional in HCT-116 cells and responded to endogenous levels of RAS activation with higher activation in HCT-116 cells with KRAS[G13D] than in the knock-out cells. Further, loss-of-function mutations in RBDCRD decreased activation (*Figure 6a*). However, the dynamic range was only threefold (*Figure 6—figure supplement 1*). Therefore, we leveraged the modularity of the circuit design to improve selectivity for target HCT-116 cells. We found that the amplified AND-gate circuits with PY2 response element, F.L.T transactivation domain, and RBDCRD as a RBD were most effective in distinguishing HCT-116[WT] from HCT-116[k.o.]. (*Figure 6b-d*). In contrast to what we saw in HEK293 overexpressing RAS (*Figure 5d*), the 'AND-gate' RAS-targeting circuits do not generate higher output than the EF1a-driven, binding-triggered RAS sensor in HCT-116. Instead, the improved dynamic range results from decreased leakiness in HCT-116[k.o.]. Only with a 3 x rigid linker in RBDCRD-NarX fusion did the PY2_NarL-F.L.T circuit show higher output expression compared to the EF1a-driven, binding-triggered RAS sensor. However, while the circuit with the 3 x rigid linker still showed a dynamic range of 18-fold, it had a decreased dynamic range compared to the same circuit with the 6 x flexible linker (57-fold; *Figure 6e*). Taken together, this dataset demonstrates that the RAS-targeting circuits are functional in cancer cells and are able to respond to endogenous levels of mutated RAS. While there are differences between the model systems, we can leverage the availability of multiple circuit parts to adapt the circuits to specific target and off-target cells to improve selectivity.

## RAS circuits can generate selective output in RAS-driven cancer cells

To specifically target RAS-driven cancer, the RAS-targeting circuits need to show selective output expression in cancer cells with mutations over-activating RAS (RAS[MUT]), while maintaining minimal expression in cells without such mutations (RAS[WT]). Testing the most promising RAS-targeting circuits in 12 cancer cell lines showed that all circuits exhibited significantly higher output expression in RAS[MUT] cells (*Figure 7a*). The PY2_NarX&NarL-F.L.T. circuit had the highest response rate among the RAS[MUT] cells but displayed slightly increased background activation in RAS[WT] cells. This was particularly notable in HT-29, a cell line harboring a BRAF mutation (*Bairoch, 2018*) and thus representing a RAS[WT] cell line but with an over-activated MAPK pathway. HT-29 did not show an elevated background in any of the other circuits, indicating a clean AND-gate behavior between the RAS sensor and the MAPK response element for the circuits that express only NarL-F.L.T. with the response elements (*Figure 7a*).

The output expression of the RAS circuits correlated with the direct expression of a fluorescent protein via the MAPK response elements, although not perfectly (*Figure 7—figure supplement 1*). This indicates that, while not the only factor, the response elements are important for the differential activation between the cell lines.

Changing the response element that regulates NarL-F.L.T. expression in the RAS circuit altered which RAS[MUT] cells showed output expression. While some RAS[MUT] cell lines, such as HCT-116 and K562, had higher output expression than all RAS[WT] cells with all tested circuits, other RAS[MUT] cell lines only showed higher expression with certain circuits, indicating that not all response elements work equally well in all RAS-driven cancer cell lines (*Figure 7a* and *Figure 7—figure supplement 2*). The availability of different response elements enabled us to identify for each RAS[MUT] cell line a functional RAS circuit with higher activation in these cells than in all RAS[WT] cell lines (*Figure 7b*). For example, while the PY2_NarL-F.L.T. circuit did not show higher activation in AsPC-1 than in the RAS[WT] cell lines (*Figure 7—figure supplement 2a*), using SRE instead of PY2 led to a 26-fold higher output expression than in the RAS[WT] cell lines (*Figure 7b*). This demonstrates that the response elements allow adapting the RAS circuits to the targeted cancer cell type.

## RAS circuits can kill RAS-driven cancer cells

To bring RAS-targeting circuits closer to therapeutic application, we replaced the fluorescent reporter with a clinically relevant output protein: a herpes simplex virus thymidine kinase variant (HSV-TK; *Preuss et al., 2010*). HSV-TK functions as a suicide gene by converting the non-toxic prodrug ganciclovir (GCV) into a cytotoxic triphosphate derivative (*Angelici et al., 2021*).

RAS-targeting circuits expressing HSV-TK induced robust cell death in KRAS-mutated HCT-116 cells after treatment with 50 µM GCV (*Figure 8a*). Compared to the non-toxic GFP-output control (GFP-circuit) condition, where cells reached full confluence, EF1a_RAS-Sensor_F.L.T. and PY2_NarL-F.L.T. reduced confluence at 180 h 1.8- and 1.7-fold, respectively. The positive control (EF1a_HSV-TK) reduced confluence 2.9-fold. Looking only at transfected cells, the effect was even more pronounced with EF1a_RAS-Sensor_F.L.T. and PY2_NarL-F.L.T. showing a 2- and 2.6-fold lower fluorescence than the GFP-circuit, respectively (*Figure 8b*). Microscopy at 180 h confirmed substantial killing. While GFP-circuit wells showed dense monolayers, those transfected with RAS-targeting circuits or EF1a_HSV-TK contained debris and rounded, aggregated cells, indicating low viability (*Figure 8c*). These results show that HSV-TK enables the RAS-targeting circuits to efficiently kill RAS-mutant HCT-116 cells, including non-transfected neighbors likely via a bystander effect (*Mesnil and Yamasaki, 2000*).

Next, we tested the circuits in Igrov-1, the RAS wild-type line that previously showed the highest activation among RAS[WT] cells for both circuits (*Figure 7a*). RAS-targeting circuits with HSV-TK did not prevent growth in Igrov-1. Following 50 µM GCV treatment, both transfected and total cell populations continued to grow (*Figure 8d&e*). In contrast, constitutive HSV-TK expression inhibited growth of transfected cells (*Figure 8d*), although without a significant effect on bystander cells (*Figure 8e*), likely due to lower transfection efficiency than HCT-116. Compared to the non-toxic GFP-circuit, EF1a_HSV-TK caused a fourfold reduction in confluence of transfected cells, while both RAS circuits showed a non-significant, 1.1-fold reduction of growth (*Figure 8d* and microscopy in *Figure 8f*). This suggests low toxicity of the RAS circuits in these RAS[WT] cells. However, it is important to note the lower transfection efficiency of Igrov-1 (*Figure 8d*), compared to HCT-116 (*Figure 8a*).

To assess the effect of transfection efficiency, we tested SW620 – a RAS[MUT] cell line with lower transfection efficiency than HCT-116 – and HCT-116 transfected with only half the DNA dose. Lower circuit dose or transfection efficiency also showed killing of RAS-driven cancer cells, but a less pronounced effect on bystander cells (*Figure 8g–l*). SW620 and Igrov-1 had similar transfection efficiency (*Figure 8—figure supplement 1a*) but distinct killing curves with SW620 showing a decrease and Igrov-1 showing an increase in confluence of transfect cells (*Figure 8d and g*), supporting the notion of selective cytotoxicity in cells with mutated RAS. However, the differences observed in transfection efficiency (*Figure 8—figure supplement 1a–b*), growth characteristics (*Figure 8—figure supplement 1c–d*), and GCV / Ef1a_HSV-TK- sensitivity (*Figure 8—figure supplement 1e–f*) may also influence RAS circuit-mediated killing, indicating that comparisons across cell lines should be interpreted cautiously.

In summary, RAS-targeting circuits expressing HSV-TK-induced cell death in transfected RAS-mutant lines (HCT-116 and SW620), while RAS wild-type Igrov-1 cells continued to grow. Although cell-line-specific differences limit final conclusions about selectivity, our data support a preferential cytotoxic effect in RAS-mutant cells. Remarkably, in KRAS[G13D]-mutated HCT-116 cells, higher DNA doses led to near-complete eradication of both transfected and neighboring cells, validating the

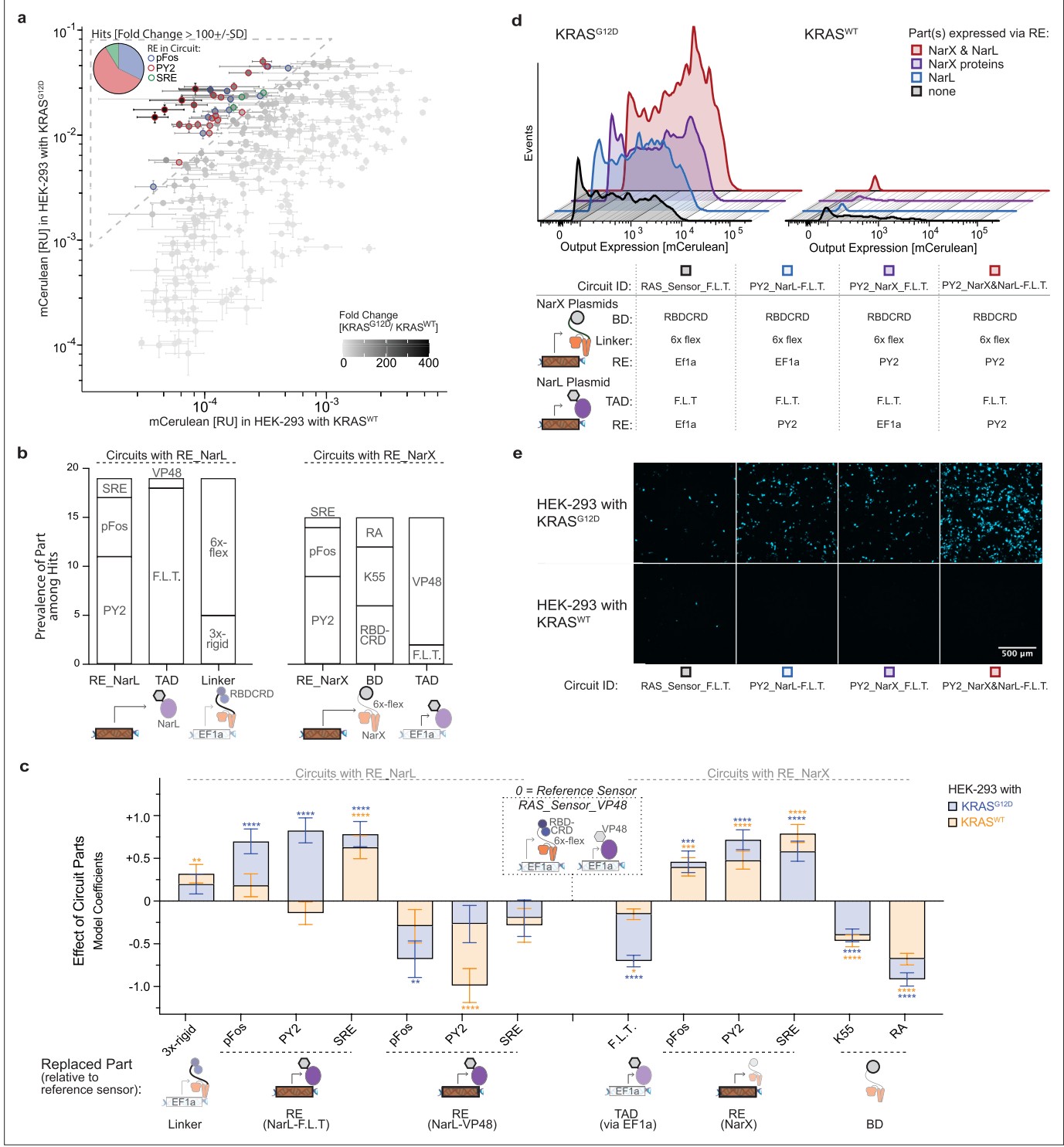

**Figure 5.** Modularity of the RAS-targeting circuits. (**a**) Screening of RAS circuit variants with different parts. mCerulean output expression in HEK293 co-transfected with 1.67 ng/well of KRAS$^{G12D}$ versus KRAS$^{WT}$ representing the ON- versus OFF-state of the screened circuits. The gray shading of the symbols represents the dynamic range (fold change between ON and OFF). Circuits with a high dynamic range (>100 +/- SD) are highlighted. The pie chart shows the prevalence of the response elements (RE) among the hits. (**b**) Prevalence of the circuit parts among the hits. On the left for all circuits where the RE expressed NarL (RE_NarL) and on the right where the RE expressed the NarX proteins (RE_NarX). TAD: transactivation domain, BD: binding domain. RA: Ras association domain of Rassf5 (**c**) Effect of different circuit parts on the output expression in HEK293 with KRAS$^{G12D}$ (blue) and KRAS$^{WT}$ (orange) fitted using a generalized linear regression. The EF1a expressed RAS Sensor with RBDCRD, a 6 x flexible linker in the NarX fusion proteins, and VP48 as TAD fused to NarL was set as the reference sensor. The graph shows the model coefficients, which can be interpreted as the

*Figure 5 continued*

effect on output expression when a part in the reference sensor is replaced by the part indicated on the x-axis. (**d**) Expression of all circuit parts via response elements. Fluorescence histograms of mCerulean-positive cells comparing ON- (KRAS$^{G12D}$) and OFF-state (KRAS$^{WT}$) of RAS circuits when either NarL (blue), the NarX proteins (violet), or all parts (red) are expressed via REs. The tested circuits contain RBDCRD as BD, a 6 x flexible linker and F.L.T. as TAD, the parts that led to the hits with the highest dynamic range. The table below shows the parts used in each of the tested circuits. Numerical values are provided in *Figure 5—source data 1*. (**e**) Microscopy images showing the mCerulean expression of the conditions from **d**. mCerulean output expression was measured by flow cytometry and normalized to a constitutively expressed mCherry transfection control. Each circuit was measured in three biological replicates. Error bars represent +/- SD. Significance in **c** was tested using the Wald test. *$p<0.05$, **$p<0.01$, ***$p<0.001$, ****$p<0.0001$.

The online version of this article includes the following source data and figure supplement(s) for figure 5:

**Source data 1.** Numerical values of flow cytometry histograms shown in *Figure 5d*.

**Source data 2.** Description of tested regression models for *Figure 5c* to select the independent variables and assess the interaction between response element and expressed parts.

**Figure supplement 1.** Differential circuit activation of the binding domains (BD) depends on the transactivation domain fused to NarL.

**Figure supplement 2.** Effect of the RAS circuit parts: response elements (RE).

**Figure supplement 3.** Effect of the RAS circuit parts: binding domain (BD) and transactivation domain (TAD) in RE_NarX circuits.

**Figure supplement 4.** Assessment of the regression model (log-transformed model used in *Figure 5*).

potential of RAS-targeting circuits to express clinically relevant output proteins and effectively kill RAS-driven cancer cells.

## Discussion

In this study, we report the design of synthetic gene circuits to target RAS-driven cancers. We developed a set of RAS sensors with interchangeable parts that can be combined to flexibly design RAS-sensing circuits. Using our modular design, we created and characterized gene circuits advancing RAS-targeting circuits on two key performance criteria: selectivity for mutated RAS and adaptability to different target and off-target cell lines.

To date, the only existing synthetic gene circuit that directly targets RAS had a twofold dynamic range between HEK cells overexpressing KRAS$^{G12V}$ and KRAS$^{WT}$ and a fourfold dynamic range when compared to HEK cells with endogenous RAS levels (*Vlahos et al., 2022*). Our design of RAS-targeting circuit greatly improves specificity for mutant RAS. This is critical given that RAS circuits sense activated RAS-GTP, which is highly overactivated in cancers with RAS mutations (*McFall et al., 2020*) but also present in healthy cells with wild-type RAS. Since this could lead to on-target, off-tumor effects, RAS-targeting circuits are designed to sense the different activation dynamics and activation levels (*Coyle and Lim, 2016*; *Li et al., 2012*) resulting from the constitutive overactivation of RAS in cancer (*McFall et al., 2020*; *Waters and Der, 2018*). To achieve high dynamic range between cells with mutated and cells with wild-type RAS, we optimized the transfer function governing the relationship between the RAS-GTP input and sensor output. To this end, we combined binding-triggered RAS sensors and RAS-dependent MAPK transcription factor sensors into a coherent type 1 feed-forward AND-gate. In this design, MAPK sensors enhance the RAS-dependent increase in expression of the binding-triggered RAS sensor components. Thus, cells with wild-type RAS have lower levels of the RAS sensor components than cells with mutant RAS, leading to suppression of RAS-dependent leakage of output production by the binding-triggered RAS sensor. This network motif was shown to delay the onset of output expression but not the output shutdown (*Alon, 2007*). This dynamic behavior, termed sign-sensitive delay (*Alon, 2007*), acts as a noise repressor (*Pieters et al., 2021*) and persistence detector (*Alon, 2007*), which could further explain why this design improves the sensing of constitutively active mutant RAS while minimizing output expression from the more transient activation of wild-type RAS (*Coyle and Lim, 2016*; *Li et al., 2012*). Integrating multiple RAS-dependent sensors into the circuit design resulted in a strongly enhanced distinction between cells overexpressing mutant or wild-type RAS and a more than 100-fold higher dynamic range than previous circuits (*Vlahos et al., 2022*).

Validation in cancer cell lines demonstrated that the RAS-targeting can selectively express an output in cancer cells. The RAS circuits are activated by all three RAS isoforms and a variety of mutations which allowed targeting of cancer cells with diverse KRAS mutations, but also of K562 cells with a BCR-ABL fusion gene that constitutively activates RAS through Sos-1 (*Cilloni and Saglio, 2012*). A

broad targeting range is advantageous because tumors exhibit high intra- and intertumoral heterogeneity of RAS mutations (*Jeantet et al., 2016*). While broader RAS inhibitors are under development (*Oya et al., 2024*), RAS heterogeneity is a limiting factor in current KRAS[G12C] inhibitors because of resistance development due to escape variants with different RAS mutations (*Steffen et al., 2023*). The broad target range of the RAS-targeting circuits enabled us to identify a circuit for each mutant RAS cell line that was more active in those cells than in all wild-type RAS cell lines. However, the

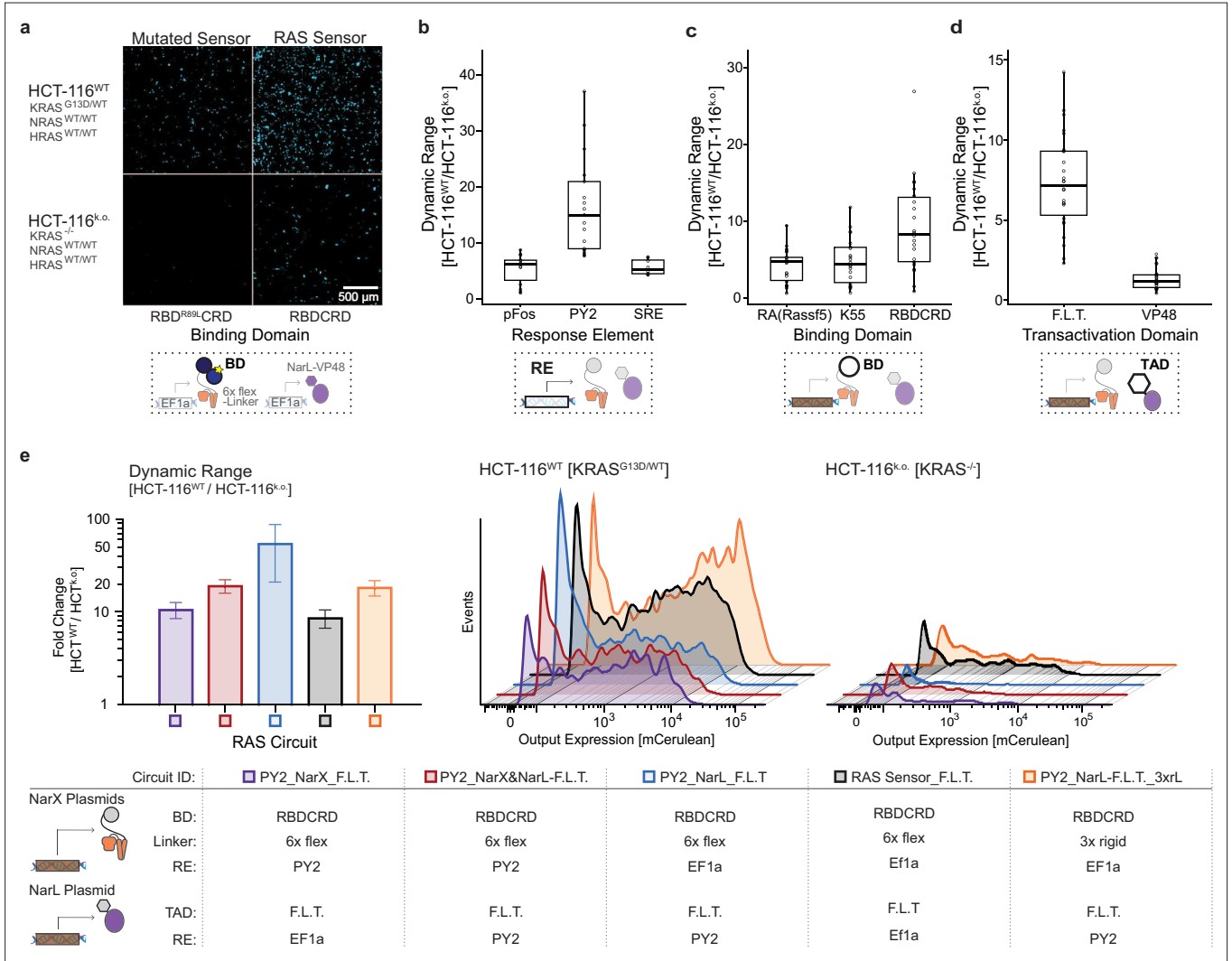

**Figure 6.** Translation into cancer cells – detection of endogenous RAS levels in HCT-116. (**a**) RAS sensor activation in colon cancer cells. Microscopy images of the mCerulean output expression in HCT-116 wild-type cells harboring a homozygous KRAS[G13D] mutation (HCT-116WT; top row) and HCT-116 KRAS knock-out cells (HCT-116k.o.; bottom row) transfected with the initial RAS sensor (right) or a RAS sensor with an R89L mutation in the Ras-binding domain (left). (**b–d**) Effect of different circuit parts in colon cancer cells. Boxplots of dynamic range of different RAS-targeting circuits grouped by the circuit parts of interest they contain. The circuit parts investigated were: the response elements in **b**, the binding domain fused to the NarX proteins in **c**, and the transactivation domain fused to NarL in **d**. Each black circle represents a different RAS circuit. (**e**) Best performing RAS-targeting circuits in colon cancer cells. The parts used in each RAS circuit are listed in the table below. The bar graph shows the dynamic range, while the fluorescence histograms show mCerulean-positive cells obtained in the On- (HCT-116WT) and Off-state (HCT-116k.o.) of the circuits. Numerical values are provided in *Figure 6—source data 1*. mCerulean output expression was measured by flow cytometry and normalized to a constitutively expressed mCherry transfection control. Dynamic range was calculated as fold change between normalized output expression in HCT-116WT and HCT-116k.o.. Each circuit was measured in three biological replicates. Error bars in **e** were calculated using error propagation rules.

The online version of this article includes the following source data and figure supplement(s) for figure 6:

**Source data 1.** Numerical values of flow cytometry histograms shown in *Figure 6e*.

**Figure supplement 1.** RAS sensor activation in HCT-116.

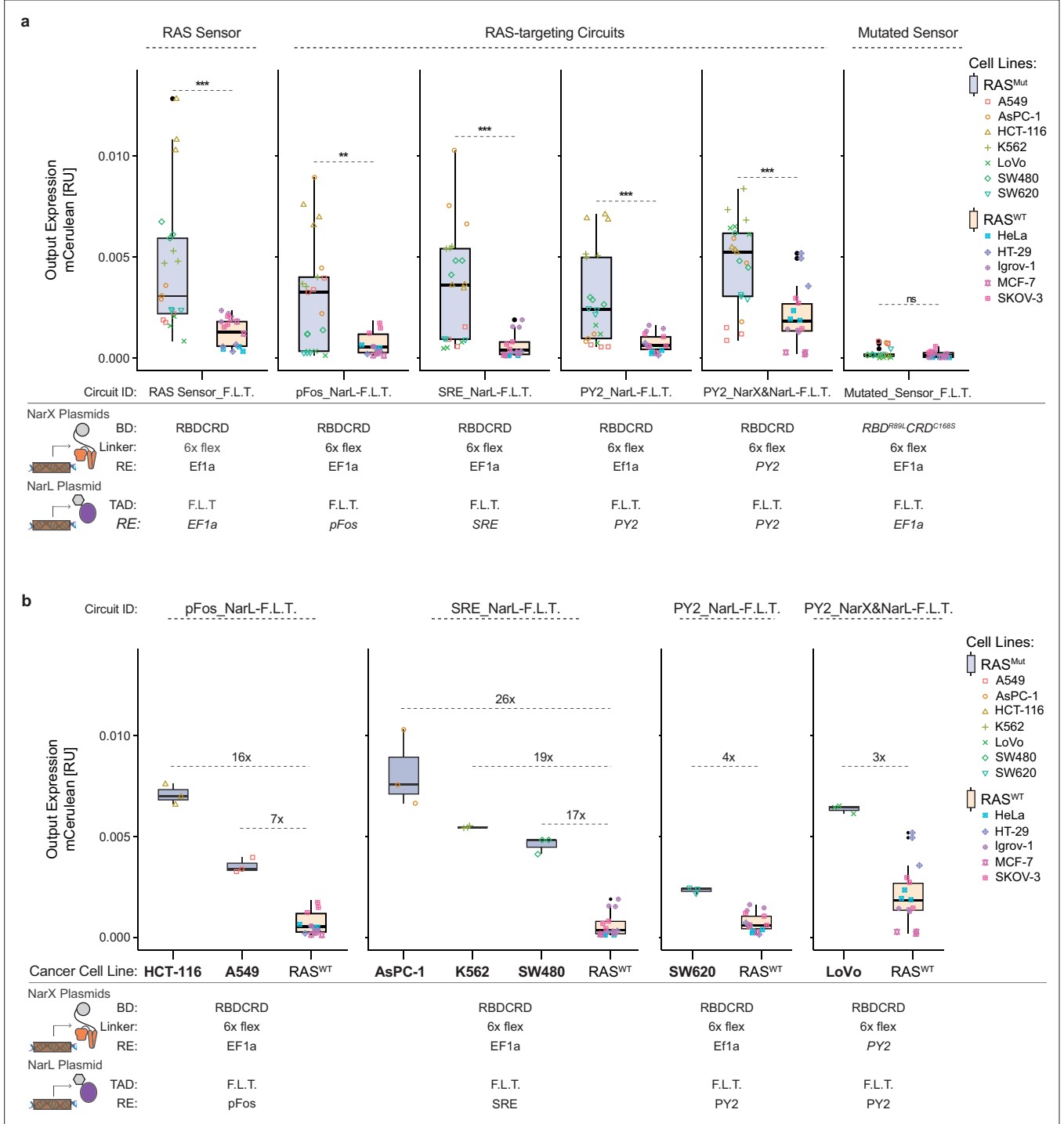

**Figure 7.** Translation into cancer cells – selectivity for RAS-driven cancer cells. (**a**) RAS-targeting circuits are classifiers for cells with mutated RAS. Output expression of RAS-targeting circuits in different cancer cell lines with (RASMUT = blue) or without (RASW$^T$ = orange) mutation leading to increased RAS activation. (**b**) Output expression of the best performing RAS-targeting circuits for each RAS$^{MUT}$ cancer cell line. Each RAS$^{MUT}$ cell line is only shown in the circuit that performed best in the respective cell line. A boxplot of all RAS$^{WT}$ cell lines is shown in each circuit, and fold changes were calculated between the mean of the individual RAS$^{MUT}$ and the mean of all RAS$^{WT}$ cell lines. The colored symbols represent biological replicates of the different cell lines. The parts used in each RAS circuit are indicated in the tables below. mCerulean output expression was measured by flow cytometry and normalized to a constitutively expressed mCherry transfection control. Each circuit was measured in three biological replicates. Significance was tested using an unpaired two-tailed Student's t-test. **p<0.01, ***p<0.001.

The online version of this article includes the following source data and figure supplement(s) for figure 7:

*Figure 7 continued on next page*

*Figure 7 continued*

**Source data 1.** Source Data used to create *Figure 7*.

**Figure supplement 1.** Correlation of activation of the MAPK sensors and the RAS-targeting circuits in individual cell lines.

**Figure supplement 2.** Comparison of RAS-targeting circuits in individual cancer cell lines with mutations overactivating RAS.

**Figure supplement 3.** Correlation of output expression and transfection efficiency for cell line testing in *Figure 7*.

heterogeneous cancer cell lines showed variable dynamic ranges, indicating a need for adaptation of the circuits to the target cells.

Additional input sensors may allow for further enhancement of the RAS-targeting circuits. Beyond the direct RAS sensors and the MAPK response elements, adding other RAS-dependent sensors, such as ERK sensors (*Ma et al., 2020*; *Stefanov et al., 2022*) or RAS-dysregulated miRNAs (*Shi et al., 2018*; *Roncarati et al., 2019*) in an AND-gate configuration could further improve specificity. Alternatively, an OR-gate configuration could reduce the risk of resistance development. Off-target effects could be minimized by including NOT-gates with inputs associated with healthy cells, such as existing p53 sensors (*Shapira et al., 2021*; *Zhan et al., 2018*) or new sensors specific for healthy cells with high RAS activation (*von Lintig et al., 2000*; *Guha et al., 1997*).

In the context of cancer therapies, synthetic gene circuits can express a variety of therapeutic proteins as output, such as proapoptotic proteins (*Xie et al., 2011*; *Gao et al., 2018*), enzymes that convert a prodrug into a cytotoxic drug (*Angelici et al., 2021*), immunotherapeutic proteins (*Nissim et al., 2017*), or even combination therapies (*Nissim et al., 2017*). We demonstrated that exchanging the output protein can be achieved by changing the coding sequence on the output plasmid. Armed with a therapeutic output, such as HSV-TK, RAS-targeting circuits can robustly kill the RAS-driven cancer cell lines HCT-116 and SW620. Simultaneously, these RAS circuits did not prevent growth in Igrov-1, suggesting low toxicity in this wild-type RAS cell line. While this indicates preferential cytotoxicity in RAS-driven cancer cells, cell-line-specific differences, such as in transfection efficiency, limit conclusions, underscoring the importance of further validation.

This further highlights another remaining challenge: delivery. The multi-plasmid-based delivery is likely difficult to implement in RAS-driven solid cancers. Thus, future efforts should aim at integrating all components on a single vector. For DNA, viral delivery is generally most efficient but has limited packaging capacity (*Shahryari et al., 2021*). Compared to EF1a, the MAPK response elements reduce the overall size of the constructs to approximately 5 kb, bringing them well within the 8 kb packaging capacity of lentiviral vectors (*Shahryari et al., 2021*) and leaving space for outputs, such as the 1.5 kb NarL-RE_HSV-TK cassette. The assembly of multiple modules on a single vector provides challenges for synthetic gene circuits, including positive and negative interactions between promoters and genes in proximity (*Dastor et al., 2018*). Using the MAPK-driven circuit versions without constitutive promoters may provide some robustness. Unintended direct output expression by neighboring MAPK response elements would still retain a certain RAS dependency, reducing the risk of constitutive, non-specific output expression in healthy cells. Nonetheless, assembling the circuit on a single vector will require careful design and rigorous validation to ensure optimal performance.

Reaching every single cancer cell will be challenging with any delivery system. We have seen that HSV-TK can also kill non-transfected, neighboring cancer cells, suggesting that outputs with bystander effect are potentially more effective. However, this effect was strongly dose-dependent, and it may require precise dosing to optimize the killing of cancer cells while minimizing toxicity. In addition to dosing the DNA amount, selection of the components used in our RAS circuits tunes the expression strength, which may allow further dosing of HSV-TK but also adaptation when using different therapeutic output proteins. Potent molecules, such as interleukin-12, will require stringent expression with low leakiness to prevent systemic toxicity (*Boumelha et al., 2023*), while less toxic molecules could benefit from stronger expression. We envision that the modularity of our system will allow us to tailor the expression profile to the therapeutic output.

In conclusion, this study provides the foundation for the design of RAS-targeting circuits. Our results confirm the feasibility of developing synthetic gene circuits that selectively target RAS-mutated cancer cells, demonstrate robust killing of certain RAS-driven cancer cells, and encourage the use of their versatility to adapt the circuits to future challenges during clinical translation. *While alternative delivery systems and validation in more realistic models will be essential*, this highlights the potential

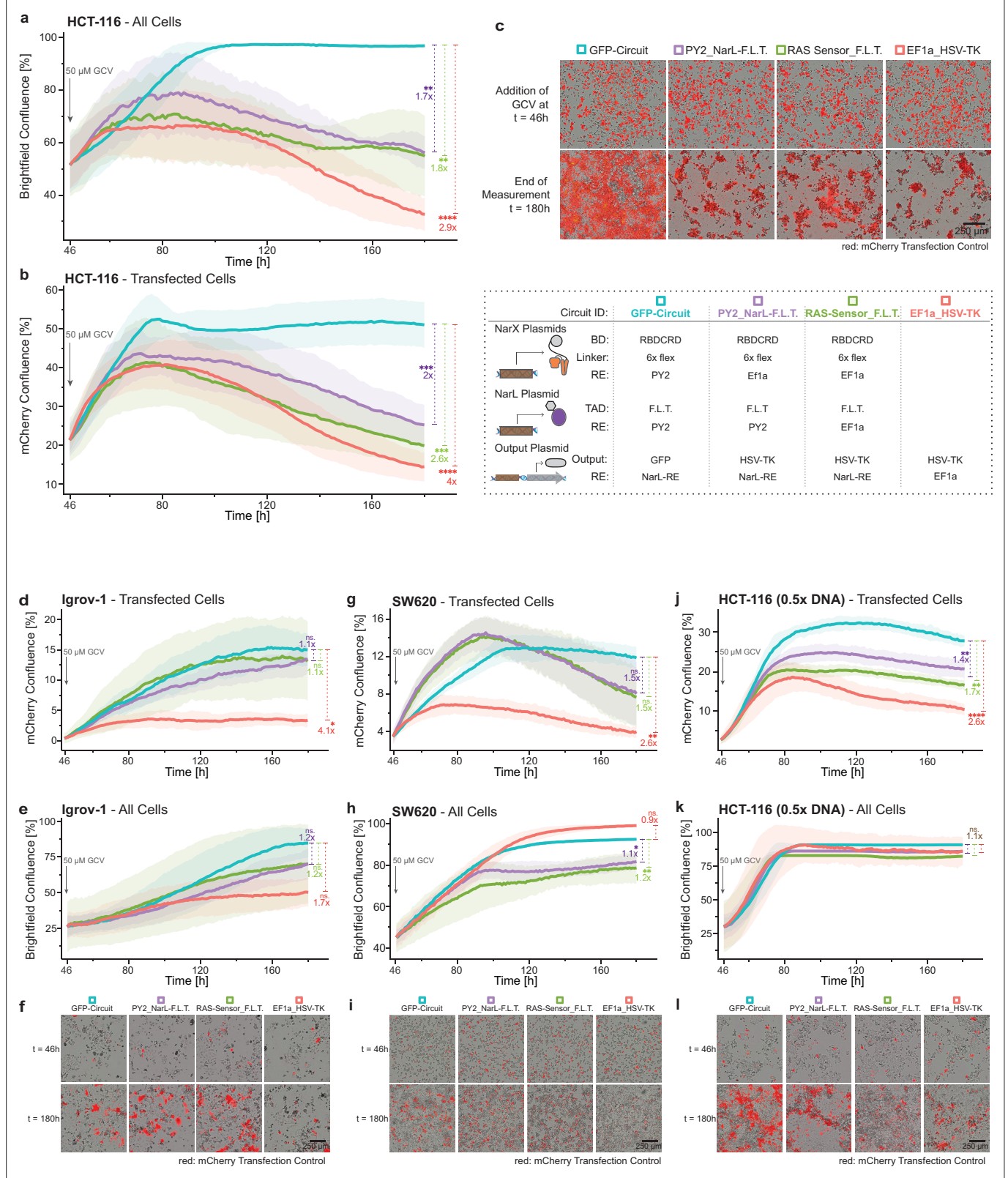

**Figure 8.** Killing of RAS-driven cancer cells. The graphs show the overall confluence, or confluence of mCherry transfection control positive cancer cells, transfected with RAS-targeting circuits that express herpes simplex virus thymidine kinase (HSV-TK) as output protein or controls over time. The used RAS circuits (purple and green) and controls are described in the dotted box, with a RAS circuit expressing GFP as output as negative control without HSV-TK (turquoise) and a EF1a-expressed constitutive HSV-TK as positive control (red). Gray arrow at t=46 h indicates the addition of the prodrug

*Figure 8 continued on next page*

*Figure 8 continued*

ganciclovir (GCV) that is activated by HSV-TK. Statistical significance and fold chances are calculated between each condition and the corresponding GFP-circuit control at t=180 h. Representative microscopy images of both brightfield and mCherry confluence at the time of ganciclovir addition (t=46) and end of the measurement (t=180) are shown for each condition and cell line. (**a**) Overall confluence in the well of KRAS^G13D-mutated HCT-116 cells. (**b**) Confluence of transfected KRAS^G13D-mutated HCT-116 cells. (**c**) Microscopy images corresponding to **a** and **b**. (**d**) Overall confluence in the well of RAS wild-type Igrov-1 cells. (**e**) Confluence of transfected RAS wild-type Igrov-1. (**f**) Microscopy images corresponding to **d** and **e**. (**g**) Overall confluence in the well of KRAS^G12V-mutated SW620 cells. (**h**) Confluence of transfected KRAS^G12V-mutated SW620 cells. (**i**) Microscopy images corresponding to **g** and **h**. (**j**) Overall confluence in the well of KRAS^G13D-mutated HCT-116 cells transfected with lower (0.5 x) amounts of the circuits. (**k**) Confluence of transfection control positive KRAS^G13D-mutated HCT-116 cells transfected with lower (0.5 x) amounts of the circuits. (**l**) Microscopy images corresponding to **j** and **k**. Confluence was quantified from microscopy images. Mean confluence was calculated from biological triplicates and background adjusted to the EF1a_HSV-TK condition of the same cell line at t=46 h. Standard deviation is shown as ribbons. Significance was tested using an ordinary one-way ANOVA with Dunnett's multiple comparison test. ns = non-significant, *$p<0.05$, **$p<0.01$, ***$p<0.001$, ****$p<0.0001$.

The online version of this article includes the following source data and figure supplement(s) for figure 8:

**Source data 1.** File with brightfield imaging source data used in *Figure 8*.

**Source data 2.** File with fluorescent (mCherry) imaging source data used in *Figure 8*.

**Figure supplement 1.** Differences between cell lines and transfection amount in killing assays.

of RAS-targeting circuits as a new therapeutic strategy that could reshape therapies against RAS-driven cancer.

## Methods

### Plasmid construction

Plasmids were cloned using standard cloning techniques, such as Gibson, GoldenGate assembly, or restriction enzyme cloning. DNA fragments were ordered from TWIST Biosciences or Genewiz (Azenta Life Sciences). Enzymes were purchased from New England Biolabs and Thermo Fisher Scientific. The sequences of all used plasmids are listed in *Supplementary file 1* and the sequences of the individual RAS circuit parts in *Supplementary file 2*.

### Cell culture

All cell lines used were cultured at 37 °C, 5% $CO_2$ in the medium suggested by the provider with 10% FBS and 1% penicillin/streptomycin solution. Cells were passaged before reaching confluency 1–3 times per week, depending on experimental plans. Details on all cell lines, such as provider, medium, and splitting ratios are listed in *Supplementary file 3*. Mycoplasma detection was performed using the protocol from the PCR Mycoplasma test kit (Promokine, Cat#PK-CA91-1024). In brief: supernatant from >80% confluent cell cultures was centrifuged (14,000 × $g$, 15 min), resuspended in water and heated (95 °C, 10 min), before performing a PCR with mycoplasma-specific primers: PR1843 (cgcc tgagtagtacgtwcgc), PR1844 (tgcctgrgtagtacattcgc), PR1845 (cgcctgagtagtatgctcgc), PR1846 (cgcc tgggtagtacattcgc), PR1847 (gcggtgtgtacaaracccga), and PR1848 (gcggtgtgtacaaaccccga). As a positive control, PR0673 (tcccacaacgaggactacac) and PR0674 (cgagtcagtgagcgaggaag) were used. Cell line identity was confirmed for all cell lines using the commercial 'cell line authentication' STR profiling service provided by Microsynth (Switzerland).

### Transfections

The transfections for the RAS-GTP pulldown experiment in *Figure 1f* and the confocal microscopy experiment in *Figure 2b* were performed in six-well plates, the transfection for the microscopy experiment in *Figure 6a*, and the cell line screening in *Figure 7* in 24-well plates, and all other experiments were performed in 96-well plates. The difference in plate size was accounted for by scaling the amount of transfected DNA to the number of seeded cells. The cells were seeded 24 hr before transfection in 100 μL of medium (500 μL for 24-well; 2.5 mL for six-well plates). Endotoxin-free (ZR DNA Prep Kit, Zymo Research, cat.no. D4201 and D4212) plasmids were mixed according to the experimental layout (*Supplementary files 4 and 5*), Opti-MEM (Thermo Fisher Scientific) was added to reach a volume of 30 μL (50 μL in 24-well plates; 500 μL in six-well plates). Transfection reagents were mixed with Opti-MEM to reach a volume of 20 μL (50 μL in 24-well plates; 500 μL in six-well plates). After incubation for at least 5 min, the mixture was added to the DNA samples, gently vortexed, spun down, and

**Table 1.** Excitation filter, dichroic mirror, and emission filter wavelengths used during fluorescence microscopy.

| Fluorochrome | Excitation filter [nm] | Dichroic mirror [nm] | Emission filter [nm] | Exposure times [ms] |
| --- | --- | --- | --- | --- |
| mCerulean | 438/24 | 458 | 483/32 | 500 |
| mCherry / mScarlet | 562/40 | 593 | 624/40 | 300 |
| SBFP2 | 370/36 | 495 | 520/35 | 300 |

then incubated at room temperature for at least 20 min before addition to the cells. The seeding and transfection conditions for each cell line are shown in *Supplementary file 6* – Table A for the cell line screening in *Figure 7* and in *Supplementary file 6* – Table B for all other experiments. To minimize the effect of differential evaporation in 96-well plates, only the inner 60 wells were used for samples, while the outer wells were filled with PBS.

For the experiments comparing different cell lines (*Figures 6–8d-l*), the DNA amount was optimized to achieve more similar transfection efficiencies, as listed in *Supplementary file 6*. In the cancer cell line screening, we adjusted the DNA amount to 1.5 x for cell lines with low transfection efficiencies (<20%), 1 x DNA for cell lines with moderate efficiency (20–50%), and 0.5 x DNA for cell lines with high efficiency (>50%).

## Structure prediction

The protein structure of the RBDCRD-linker-NarX fusion proteins was predicted with AlphaFold2 from the amino acid sequence using the Latch Console platform (LatchBio). Using PyMOL (Version 2.5.4, Schrödinger, LLC.), we aligned the NMR-derived structure of two KRAS-RBDCRD dimers tethered to a nanodisc (worldwide Protein Data Bank, PDB accession code: 6PTS *Fang et al., 2020*) with the NMR structure of a KRAS4B-GTP homodimer on a lipid bilayer nanodisc (PDB accession code: 6W4E *Lee et al., 2020*). The predicted structure of two RBDCRD-linker-NarX proteins was then aligned to each of the RBDCRD structures. The flexible parts of the RBDCRD and the linkers were rotated to bring the two NarX domains into proximity.

## Fluorescence microscopy

Microscopy pictures were taken 36–48 hr after transfection using a Nikon Eclipse Ti inverted microscope equipped with a Nikon Intensilight C-HGFI fiber illuminator, Semrock filter cubes (IDEX Health & Science), a 10 x objective, and a Hamamatsu C10600 ORCA-R2 digital camera. Excitation filters, dichroic mirrors, emission filters, and exposure times are summarized in *Table 1*. The look-up table (LUT) values were adjusted for ideal contrast and kept constant within an experiment.

## Flow cytometry

36–48 hr after transfection, we prepared the cells for flow cytometry analysis by removing the medium and adding 70–100 µL of Accuatase (Gibco, Thermo Fisher Scientific, cat.no. #A1110501). The cells were incubated for 15–30 min at room temperature and then kept on ice before measurement using a BD LSRFortessa with a high-throughput screening device. To avoid potential cell damage and minimize time on ice, the plates were prepared consecutively, right before analysis. Excitation wavelengths, longpass filters, and emission bandpass filters were optimized to reduce crosstalk between different fluorophores and are summarized in *Table 2*. When working with 24-well plates, after removing the medium, 150–200 µL of Accutase was used to detach the cells. After 15–30 min of incubation, the cells

**Table 2.** Excitation laser, long pass filter, and emission filter wavelengths used during flow cytometry.

| Fluorochrome | Excitation laser [nm] | Long pass filter [nm] | Emission filter [nm] |
| --- | --- | --- | --- |
| mCerulean | 445 | - | 473/10 |
| SYFP2/mVenus | 488 | 505 | 530/11 |
| mCherry / mScarlet | 561 | 600 | 610/20 |
| SBFP2 | 405 | - | 445/15 |

were detached by gentle pipetting, and the complete cell suspension was transferred to a 96-well plate for analysis using the HTS device.

## RAS-GTP pulldown ELISA

RAS-GTP levels were measured using a Ras Activation ELISA assay kit (Merck Millipore cat.no. #17–497) according to the manual. HEK293 cells were seeded in six-well plates and transfected as described in the 'Transfection' section. Cells were lysed using 250 µL of the provided lysis buffer with added Halt Protease Inhibitor Cocktail (Thermo Fisher Scientific, cat.no. #89900). Samples were snap frozen in liquid nitrogen and stored at –80 °C. An aliquot was used to quantify the protein amount in the cell lysates using a Pierce BCA Protein Assay Kit (Thermo Fisher Scientific, cat.no #23227). The next day, 100 µg of each sample was used for the ELISA. The anti-RAS antibody (Merck Millipore cat.no. #17–497; part no.2006992) was provided in the ELISA kit. Chemiluminescence was measured using a Tecan Spark Multimode Microplate Reader and measured 20 min after the addition of the substrate with an integration time of 1 s.

## Confocal microscopy for quantification of membrane binding

Cells were seeded in a 24-well plate and transfected as described in the 'Transfections' section. After 36 hr, we detached the cells using 0.25% trypsin and re-seeded 30,000 cells in 200 µL medium into an eight-well glass-bottom plate to have sparsely distributed cells suitable for membrane detection and image analysis. The cells were incubated at 37 °C, 5% $CO2$ to reattach. After 4 hr, we stained the membrane of the cells with the CellBrite Steady 685 Membrane Staining Kit (20 µL of 1:100 dilution; Biotium cat.no. #30,109 T) and the nuclei with 5 µL NucBlue Live Cell Stain (Thermo Fisher Scientific, cat.no. R37605). The wells were imaged using a Falcon SP8 confocal microscope (Leica Microsystems) with a 20 x objective. SYFP2 was measured using an excitation laser at 524 nm and 540–600 nm bandpass filter, NucBlue was measured using the 405 nm laser and 415–460 nm bandpass filter, CellBrite was measured using a 670 nm excitation laser and a 750–800 nm bandpass filter.

## Image analysis for quantification of membrane binding

Confocal images were analyzed with Python v3.10 using the Scikit-Image v0.19.2, SciPy v1.8.1, NumPy v1.22.4, Pandas v1.4.2, OpenCV v4.5.0 and custom packages built upon these libraries. Cells were segmented based on the membrane (CellBrite Steady 685) and nuclear (NucBlue) signals. The nuclear signals were used to assign the membrane signal to the individual cell. First, each nucleus was identified as a single cell. Second, the membrane signal was assigned to the nearest nucleus and used to create a mask for the membrane of each cell. Third, the membrane masks were post-processed to exclude non-closed objects, such as cell debris or cells with incomplete membrane staining as well as cells with more than one nucleus per cell. Fourth, the membrane masks were filled to additionally obtain the full cell masks. Fifth, for signal analysis, the total fluorescence in the SYFP2 channel was calculated for both the membrane and the full cell mask of each cell. Finally, the ratio between the membrane and the total cell signal was calculated for each cell. We used untransfected cells stained with NucBlue and CellBright Steady 685 to assess background fluorescence. For the analysis, we included all transfected cells with total cell fluorescence above this background. The code is available on GitHub (copy archived at *Senn and Aaron, 2026*).

## Regression model

The effect of different circuit parts on the output expression in HEK-293 with KRAS[G12D], in HEK-293 with KRAS[WT] as well as on the dynamic range (KRAS[G12D]/KRAS[WT]) was modeled using a Gaussian generalized linear model in R Studio v2023.06.0+421. Because the datasets were skewed toward low values, resulting in a loss of model accuracy at low values, a log transformation of all values was performed before running the models. Linearity and homoscedasticity assumptions were assessed using residual plots (*Figure 5—figure supplement 4*). The goodness of fit was evaluated by comparing the fitted and measured values and calculating Pearson ($R^2$) and Spearman correlations using the cor() function in R Studio (*Figure 5—figure supplement 4*). For the selection of the independent variables, we tested three models assuming different interactions between the response elements and the parts they express. (*Figure 5—source data 2*). The model with the best correlation (Model 3) was selected to analyze the effect of the different circuit parts in *Figure 5c*. In addition to the primary variables of

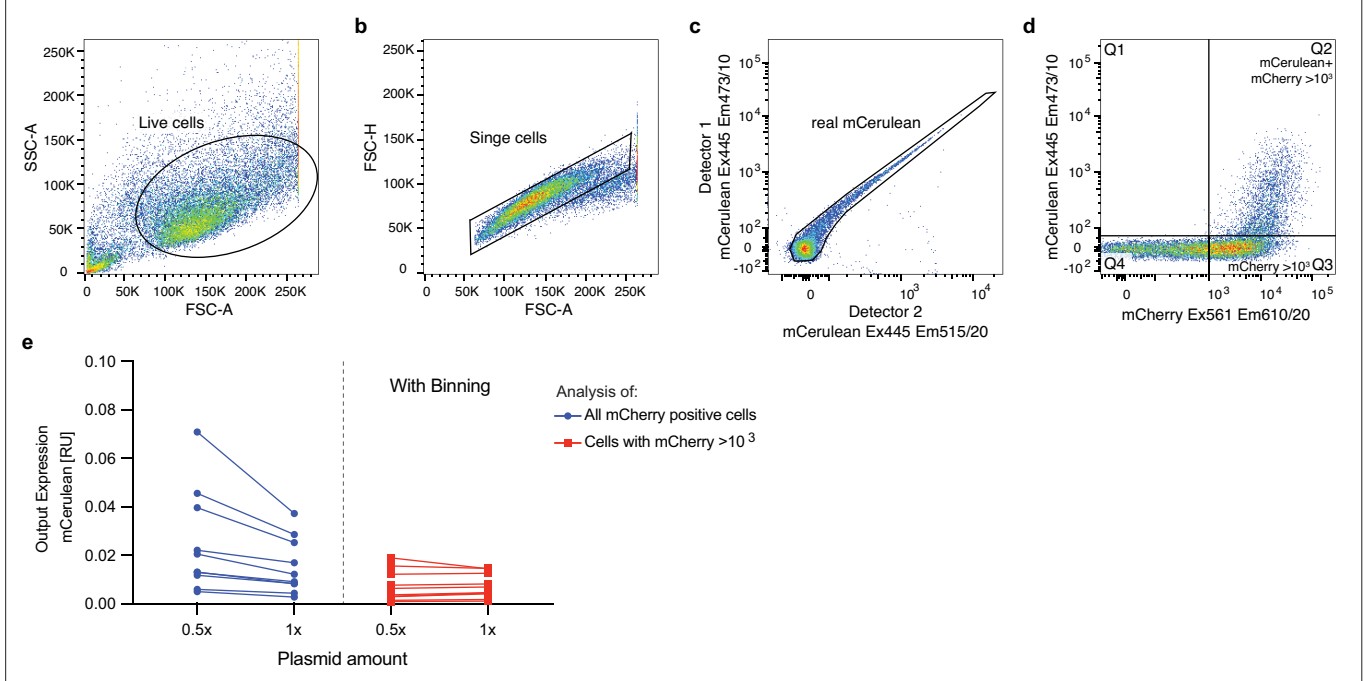

**Figure 9.** Illustration of gating in flow cytometry analyses. Gating example in HEK293 cells transfected with a mCerulean expressing RAS circuit and mCherry transfection control (**a**) Live cells are gated by plotting all events on a side scatter area (SSC-A) versus forward scatter area (FSC-A) density plot. (**b**) Single cells are gated by plotting FSC-A versus FSC-height (FSC-H). (**c**) Real mCerulean signals are separated from false positive signals by plotting the mCerulean signal measured with two independent detectors against each other. Both measure the signal from the 445 nm excitation laser, only exiting mCerulean and not mCherry. Both have an emission filter that can strongly detect mCerulean. Detector one measures mCerulean with a 473/11 emission filter, while detector two measures mCerulean with a 515/20 emission filter. Only real mCerulean signals where the signal correlates between the detectors are inside the gate. (**d**) Plot showing mCerulean output signal (excitation at 445 nm, emission filter 473/10) versus mCherry transfection control signal (excitation at 561 nm, emission filter 610/20). mCerulean output expression is calculated by multiplying the frequency of parent * mean of Q2, which represents mCerulean positive cells with high transfection (mCherry higher than $10^3$). The mCherry signal is calculated by multiplying the frequency of parent * mean of all cells with mCherry higher than $10^3$ (Q2+Q3). 103 was chosen as the threshold for transfection efficiency as cells below this rarely show circuit activation. (**e**) Binning makes the normalized Output signal less sensitive to different transfection amounts. Change in mCerulean output expression in HCT-116 cells transfected with 1 x or 0.5 x the plasmid amount of various RAS circuits to simulate different transfection efficiencies. All mCherry-positive cells were analyzed on the left (blue), while on the right the cells were binned for high transfection efficiency, and only cells with mCherry signal $>10^3$ were analyzed (red).

interest, we considered potential covariates such as experimental batch, transfection efficiency, and amount of NarX and NarL plasmids. While the experimental batch did not affect the correlation of the model, transfection efficiency and plasmid amounts were included in the regression to ensure the robustness of the model.

## Analysis of flow cytometry data

Flow cytometry data analysis was performed using FlowJo software v10.8.0 (BD Life Sciences). The gating strategy is shown in (**Figure 9a–d**). When multiple cell lines with different transfection efficiencies were used, the cells were binned using the expression of the mCherry transfection control to include only cells with high transfection levels $>10^3$ (**Figure 9d–e**). For testing of different cell lines in **Figure 7**, we further validated that there was no direct correlation between transfection efficiency and normalized output in our experimental data (**Figure 7—figure supplement 3**). Absolute units (AU) of fluorescence were calculated by multiplying the number of positive cells (frequency of parent) with the mean expression of the fluorophore:

$$\textit{Frequency of Parent}_{Fluorophor} * \textit{Mean}_{Fluorophor}$$

Relative units (RU) of fluorescence were calculated by dividing the AU of the fluorophore of interest by the AU of the transfection control:

$$Fluorescence\,[RU] = \frac{Frequency\,of\,Parent_{Fluorophor\,of\,Interest} * Mean_{Fluorophor\,of\,Interest}}{Frequency\,of\,Parent_{Transfection\,Control} * Mean_{Transfection\,Control}}$$

For the dynamic range, the fold change between the target cell line (HEK$^{G12D}$ or HCT-116$^{WT}$; ON-state) and off-target cell line (HEK$^{G12D}$ or HCT-116$^{WT}$; OFF-state) was calculated. The error bars were calculated using error propagation rules:

$$Propagated\,Uncertainty = Fold\,Change * \sqrt{\left(\frac{SD_{ON-state}}{Mean_{ON-state}}\right)^2 + \left(\frac{SD_{OFF-state}}{Mean_{OFF-state}}\right)^2}$$

The flow cytometry histograms in *Figures 5d and 6e* show all mCerulean-positive cells concatenated from the three biological replicates. The numerical values of the cell number, mean, and frequency of parent can be found in *Figure 5—source data 1* and *Figure 6—source data 1* respectively.

## Analysis and visualization of screening data

Data analysis and plotting of the data from the screening of different circuit parts (*Figure 5*) and the screening of cancer cell lines (*Figure 7*) were performed using R studio v2023.06.0+421. The data was imported from FlowJo analysis and labeled with the corresponding sample descriptors before calculating the mean and SD for the biological replicates. Plots were generated using ggplot2 v3.4.3 and cowplot v1.1.1. Large language models (ChatGPT3.5, ChatGPT4.0; OpenAI) were used to facilitate code writing.

## HSV-TK killing assays

Cells were transfected as described under transfection, except that only half the number of cells were seeded, to adjust for the long duration of the assay. 46 hr after seeding (ca. 36 hr after transfection), 100 µL of medium with GCV (Sigma, cat.no. SML2346-1ML) or was added to reach a final concentration of 50 µM. Brightfield and mCherry confluences of the cells were continuously imaged over the time course of the assay using the xCelligence eSight Real-Time Cell Analysis (Agilent, USA, CA). The Agilent RTCA eSight software v.1.3.2 was used to create brightfield and red fluorescence segmentation masks and quantify the confluence. Segmentation parameters were selected for each cell line to optimize for cell size and background-to-cell contrast (*Supplementary file 7*). Quantified confluence data was exported and analyzed using R Studio v2023.06.0+421. To adjust for differences in confluence before addition of GCV between the conditions, the confluence shown in *Figure 8* was background adjusted to the EF1a_HSV-TK condition of the same cell line at t=46h:

$$Background\,adjusted\,Mean_{t=n} = Mean_{t=n} + \left(Mean_{HSVTK\,at\,t=46} - Mean_{t=46}\right)$$

## Statistical analysis

In experiments comparing two groups, unpaired two-tailed Student's t-tests were used to assess significance (*Figures 1c and 7a*, *Figure 4—figure supplement 3d–l* and *Figure 7—figure supplement 3*). When comparing three or more groups, an ordinary one-way ANOVA was used followed by a Dunnett's multiple comparison test when all groups were compared with a control column (*Figures 2 and 8*, *Figure 5—figure supplement 2* and *Figure 5—figure supplement 3a-c*) or a Tukey's test when all columns were compared (*Figure 8*, *Figure 1—figure supplement 2b-f* and *Figure 5—figure supplement 1a–b*). Significance in the regression model (*Figure 5c*) was tested using a Wald test. Data was considered statistically significant at a p-value below 0.05. The number of replicates is provided in the figure captions. Each replicate was taken from a distinct sample. Apart from the large screening experiments (*Figures 5 and 6*), data is representative of at least two experiments. The p-values, F-values, t-values, and degrees of freedom (df) from all statistical comparisons are provided in *Supplementary file 8*. Statistical analyses were performed using Prism 10 (GraphPad) or R Studio (v2023.06.0+421).

## Acknowledgements

This work was supported by the National Center of Competence in Research Molecular Systems Engineering (NCCR-MSE) and ETH Zurich core funding. We thank M Di Tacchio, C Cavallini, A Gumienny,

E Montani, and A Ponti for assistance with flow cytometry and image acquisition as well as analysis. We also thank B Treutlein for discussions, proofreading, and providing the lab infrastructure during the last part of the project. We thank D Schweingruber, F Trick, V Cheras, and S Seidel for proofreading. Thank you to M Lampis and V Cheras for support during revisions. Lastly, we would like to thank A Abraham, J Jaekel, J Schreiber, M Dastor, P Müller-Thümen, and all Benenson and Treutlein lab members for discussions.

## Additional information

### Competing interests
Yaakov Benenson: Shareholder and an employee of Pattern Biosciences. The other authors declare that no competing interests exist.

### Funding

| Funder | Grant reference number | Author |
|---|---|---|
| National Center of Competence in Research - Molecular Systems Engineering | | Yaakov Benenson |
| ETH Zurich | Core Funding | Yaakov Benenson |

The funders had no role in study design, data collection and interpretation, or the decision to submit the work for publication.

### Author contributions
Gabriel Valentin Senn, Conceptualization, Data curation, Formal analysis, Validation, Investigation, Visualization, Methodology, Writing – original draft, Writing – review and editing; Leon Nissen, Investigation; Yaakov Benenson, Conceptualization, Resources, Supervision, Funding acquisition, Writing – original draft, Project administration, Writing – review and editing

### Author ORCIDs
Gabriel Valentin Senn ⓘ https://orcid.org/0009-0002-7123-2899
Leon Nissen ⓘ https://orcid.org/0009-0009-3272-834X
Yaakov Benenson ⓘ https://orcid.org/0000-0003-1880-6507

Reviewer #1 (Public review): https://doi.org/10.7554/eLife.104320.3.sa1
Reviewer #2 (Public review): https://doi.org/10.7554/eLife.104320.3.sa2
Reviewer #3 (Public review): https://doi.org/10.7554/eLife.104320.3.sa3
Author response https://doi.org/10.7554/eLife.104320.3.sa4

## Additional files

### Supplementary files
Supplementary file 1. Overview & sequences of all used plasmids.

Supplementary file 2. Sequences of all circuit parts.

Supplementary file 3. Cell line & culturing details.

Supplementary file 4. Transfection & experimental layout details.

Supplementary file 5. Transfection details for screening of circuit parts in *Figure 5a–c*.

Supplementary file 6. Seeding and Transfection conditions.

Supplementary file 7. Segmentation parameters for image analysis using Agilent eSight in *Figure 8*.

Supplementary file 8. Overview of used statistical tests.

MDAR checklist

## Data availability

Full sequences of all plasmids and fragments used are provided in *Supplementary files 1 and 2*. Experimental set-up for all transfections is provided in the *Supplementary files 4 and 5*. *Supplementary file 5* contains the data used for the regression model in *Figure 5*. Data for *Figures 7 and 8* are available in *Figure 7—source data 1*, *Figure 8—source data 1* and *Figure 8—source data 2*, respectively. The formula and independent variables used for the regression are shown in *Figure 5—source data 1*. The statistical tests and analysis used for each figure are provided in the legends. Code used to analyze the membrane localization of the RAS sensor in *Figure 2b* and R scripts for plotting of *Figures 7 and 8* are available on Github (https://github.com/gabsenn/RAS-targeting_Gene_Circuits; copy archived at *Senn and Aaron, 2026*).

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
