## [Editor Report · eLife Assessment]

This **important** study demonstrates the potential of synthetic gene circuits to detect and target aberrant RAS activity in cancer cell lines. The circuit design is novel and the evidence supporting the claims is **convincing**. As a proof-of-concept, this will be of broad interest to researchers in synthetic biology and therapeutics development, while future work will be required to help translate this technology toward clinical applications in cancer therapeutics and address potential limitations of the strategy.

---

## [Referee Report · Reviewer #1 (Public review)]

Summary:

The manuscript presents a comprehensive study on the developing synthetic gene circuits targeting mutant RAS expressing cells. The aim of this study is to use these RAS targeting circuits as cancer cell classifiers and enable the selective expression of an output protein in correlation with RAS activity. The system is based on the bacterial two-component system NarX/NarL. A RAS-binding domain is fused to a NarX mutant either defective in the ATP binding (N509A) or the phosphorylation site (H399Q). Nanocluster formation of RAS-GTP reconstitutes an active histidine kinase sensor dimer that phosphorylates the response regulator NarL thus leading to the expression of an output protein. The integration of RAS-dependent MAPK responsive elements to express the RAS sensor components generates RAS circuits with an extended dynamic range between mutant and wild-type RAS. The selectivity of the RAS circuits is confirmed in a set of cancer cell lines expressing endogenous levels of mutant or wild-type RAS or oncogenes affecting RAS signaling upstream or downstream. Expression of the suicide gene HSV thymidine kinase as an outcome protein kills RAS-driven cancer cells demonstrating the functionality of the system.

Strengths:

This proof-of-concept study convincingly demonstrates the potential of synthetic gene circuits to target oncogenic RAS in tumor cell lines, act as RAS mutant cell classifier, and induce the killing of RAS-driven cells.

Weaknesses:

A therapeutic strategy based on of this four-plasmid system may be difficult to implement in RAS-driven solid cancers. However, potential solutions are discussed.

---

## [Referee Report · Reviewer #2 (Public review)]

The manuscript describes an interesting approach towards designing genetic circuits to sense different RAS mutants in the context of cancer therapeutics. The authors created sensors for mutant RAS and incorporated feed-forward control that leverages endogenous RAS/MAPK signaling pathways in order to dramatically increase the circuits' dynamic range. The modularity of the system is explored through the individual screening of several RAS binding domains, transmembrane domains, and MAPK response elements, and the author further extensively screened different combinations of circuit components. This is an impressive synthetic biology demonstration that took it all the way to cancer cell lines. However, given the sole demonstrated output in the form of fluorescent proteins, the authors' claims related to therapeutic implications require additional empirical evidence or, otherwise, expository revision.

Major comments:

"These therapies are limited to cancers with KRASG12C mutations" is technically accurate. However, in this fast-moving field, there are examples such as MRTX1133 which holds the promise to target the very G12D mutation that is the focus of this paper. There are broader efforts too. It would help the readers better appreciate the background if the authors could update the intro to reflect the most recent landscape of RAS-targeting drugs.

Only KRASG12D was used as a model in the design and optimization work of the genetic circuits. Other mutations should be quite experimentally feasible and comparisons of the circuits' performances across different KRAS mutations would allow for stronger claims on the circuits' generalizability. Particularly, the cancer cell line used for circuit validation harbored a KRASG13D mutation. While the data presented do indeed support the circuit's "generalizability," the model systems would not have been consistent in the current set of data presented.

In Figure 2a, the text claims that "inactivation of endogenous RAS with NF1 resulted in a lower YFP/RBDCRD-NarX expression," but Figure 2a does not show a statistically significant reduction in expression of SYFP (measured by "membrane-to-total signal ratio [RU]).

The therapeutic index of the authors' systems would be better characterized by a functional payload, other than florescent proteins, that for example induce cell death, immune responses, etc.

Regarding data presented in "Mechanism of action" (Figure 2), the observations are interesting and consistent across different fluorescent reporters. However, with regard to interpretations of the underlying molecular mechanisms, it is not clear whether the different output levels in 2b, 2c, and 2d are due to the pathway as described by the authors or simply from varied expression levels of RBDCRD-NarX itself (2a) that is nonlinearly amplified by the rest of the circuit. From a practical standpoint, this caveat is not critical with respect to the signal-to-noise ratios in later parts of the paper. From a mechanistic interpretation standpoint, claims made forth in this section are not clearly substantiated. Some additional controls would be nice. For example, if the authors express NarXs that constitutively dimerize on the membrane, what would the RasG12D-responsiveness look like? Does RasG12D alter the input-output curve of NarL-RE? How would Figure 4f compare to a NaxR constitutively dimerized control that only relies on transcriptional amplification of the Ras-dependent promoters? It's also possible that these Ras could affect protein production at the post-transcriptional or even post-translational levels, which were not adequately considered.

The text claims that "in contrast to what we saw in HEK293 overexpressing RAS (Figure 5d), the "AND-gate" RAS-targeting circuits do not generate higher output than the EF1a-driven, binding-triggered RAS sensor in HCT-116. Instead, the improved dynamic range results from decreased leakiness in HCT- 116k.o." Comparing the experiment from Figure 5d, which looks at activation in KRASG12D and KRASWT, to the experiments in Figure 6b-d, which looks at activation in HCT-116WT and HCT-116KO is misleading. In Fig 5d., cells are transfected with KRASG12D and KRASWT to emulate high levels of mutant RAS and high levels of wild-type RAS. In Figures 6b-d, HCT-116WT has endogenous levels of mutant RAS, while the KCT-116KO is a knock-out cell line, and does not have mutant or WT RAS. Therefore, the improved dynamic range or "decreased leakiness in HCT-116KO" in comparison to Figure 5d. is more comparable to the NF1 condition from Figure 2, which deactivates endogenous RAS. While this may not be feasible, the most accurate comparison would have been an HCT-116KO line with KRASWT stably integrated.

We couldn't locate the citation or discussion of Figure 4d in the text. Conversely, based on the text description, Figure 6g would contain exciting results. But we couldn't find Figure 6g anywhere ... unless it was a typo and the authors meant Figure 6f, in which case the cool results in Figure S8 could use more elaboration in the main text.

Comments on revisions:

Now that the authors have extensively addressed my comments through text and additional experiments, I am supportive of its conclusions. I thank them for the rigorous updates and congratulate them on an important piece demonstrating the potential of synthetic biology circuits.

---

## [Referee Report · Reviewer #3 (Public review)]

Summary:

Mutations that result in consistent RAS activation constitute a major driver of cancer. Therefore, RAS is a favorable target for cancer therapy. However, since normal RAS activity is essential for the function of normal cells, a mechanism that differentiates aberrant RAS activity from normal one is required to avoid severe adverse effects. To this end, the authors designed and optimized a synthetic gene circuit that is induced by active RAS-GTP. The circuit components, such as RAS-GTP sensors, dimerization domains, and linkers. To enhance the circuit selectivity and dynamic range, the authors designed a synthetic promoter comprised of MAPK-responsive elements to regulate the expression of the RAS sensors, thus generating a feed-forward loop regulating the circuit components. Circuit outputs with respect to circuit design modification were characterized in standard model cell lines using basal RAS activity, active RAS mutants, and RAS inactivation.

This approach is interesting. The design is novel and could be implemented for other RAS-mediated applications. The data support the claims, and while this circuit may require further optimization for clinical application, it is an interesting proof of concept for targeting of aberrant RAS activity. I therefore recommend accepting this paper.

Strengths:

Novel circuit design, through optimization and characterization of the circuit components, solid data.

Weaknesses:

This manuscript could significantly benefit from testing the circuit performance in more realistic cell lines, such as patient-derived cells driven by RAS mutations, as well as in corresponding non-cancer cell lines with normal RAS activity. Furthermore, testing with therapeutic output proteins in vitro, and especially in vivo, would significantly strengthen the findings and claims.

Summary:

Given the revision made, I would recommend a minor revision that discusses the specificity limitations of this experimental setup.

---

## [Author Response]

The following is the authors’ response to the original reviews.

**Reviewer #1 (Public review):**
Summary:The manuscript by Senn and colleagues presents a comprehensive study on the developing synthetic gene circuits targeting mutant RAS-expressing cells. This study aims to exploit these RAS-targeting circuits as cancer cell classifiers, enabling the selective expression of an output protein in correlation with RAS activity. The system is based on the bacterial two-component system NarX/NarL. A RAS-binding domain, the RBDCRD domain of the RAS effector protein CRAF, is fused to the histidine kinase domain, which carries an inactivating amino acid exchange either in its ATP-binding site (N509A) or in its phosphorylation site (H399Q). Dimerization or nanocluster formation of RAS-GTP reconstitutes an active histidine kinase sensor dimer that phosphorylates the response regulator NarL. The phosphorylated DNA-binding protein NarL, fused to the transcription activator domain VP48, binds its responsive element and induces the expression of the output protein. In comparison to mutated RAS, the effect of the RAS activator SOS-1 and the RAS inhibitor NF1 on the sensing ability as well as the tunability of the RAS sensor were examined. A RAS targeting circuit with an AND gate was designed by expressing the RAS sensor proteins under the control of defined MAPK response elements, resulting in a large increase in the dynamic range between mutant and wild-type RAS. Finally, the RAS targeting circuits were evaluated in detail in a set of twelve cancer cell lines expressing endogenous levels of mutant or wild-type RAS or oncogenes affecting RAS signaling upstream or downstream.Strengths:This proof-of-concept study convincingly demonstrates the potential of synthetic gene circuits to target oncogenic RAS in tumor cell lines and to function, at least in part, as an RAS mutant cell classifier.Weaknesses:The use of an appropriate "therapeutic gene" might revert the oncogenic properties of RAS mutant cell lines. However, a therapeutic strategy based on this four-plasmid-based system might be difficult to implement in RAS-driven solid cancers.

Thank you for the insightful comments. We agree that the delivery of a four-plasmid system represents a major challenge for translating RAS-targeting circuits into therapeutic applications. Reducing the number of plasmids –ideally consolidating all components onto a single vector– will be critical for clinical implementation.

Viral delivery is generally the most efficient strategy for DNA-based therapies, but viral vectors have limited packaging capacities, which differ by virus type[1]. The RAS_sensor_F.L.T. circuit under the EF1α promoter requires ~7.7 kb for the sensing components alone, excluding the output gene. This exceeds the packaging limit of adeno-associated virus (AAV) and is at the upper boundary for lentiviral vectors but could potentially be accommodated by larger vectors such as γ-retroviruses, poxviruses, or herpesviruses¹. Co-transduction with dual AAVs [2] or ongoing engineering to expand packaging capacity [3] may also offer future solutions. An additional route to reduce construct size could be alternative splicing, especially given redundancy between the two NarX fusion proteins[4].

An advantage of our current architecture is that synthetic response elements replace constitutive promoters, reducing construct size. For example, the MAPK-driven PY2_NarX&NarL circuits range between 4.9 and 5.2 kb depending on the transactivation domain, bringing them within AAV packaging limits for the sensor module[5], though co-delivery of the output gene would still be necessary. For lentiviruses, this is within the packaging capacity of 8 kb^1^ and would allow for inclusion of ~3 kb output genes.

Still, assembling multiple modules onto a single vector introduces new challenges, including possible crosstalk or interference between neighboring promoters [6]. For example, placing the output gene too close to MAPK response elements may trigger unwanted MAPKdependent expression, potentially bypassing the intended AND-gate logic. Moreover, expressing three genes under separate response elements may shift expression ratios and reduce circuit functionality. Nonetheless, the absence of constitutive promoters and the RAS-dependence of MAPK response elements could provide partial robustness, since even unintended activation would still reflect RAS signaling to some extent. Further, our data (Fig. 1d) show that some deviation in component levels can be tolerated, provided all parts are sufficiently expressed. Nonetheless, assembling the circuit on a single vector will require careful design and rigorous validation to ensure optimal performance.

While addressing this is beyond the scope of the current study, we agree that future efforts should focus on vector consolidation and delivery strategies. We now include a paragraph discussing these challenges in the revised manuscript.

**Reviewer #2 (Public review):**
The manuscript describes an interesting approach towards designing genetic circuits to sense different RAS mutants in the context of cancer therapeutics. The authors created sensors for mutant RAS and incorporated feed-forward control that leverages endogenous RAS/MAPK signaling pathways in order to dramatically increase the circuits' dynamic range. The modularity of the system is explored through the individual screening of several RAS binding domains, transmembrane domains, and MAPK response elements, and the author further extensively screened different combinations of circuit components. This is an impressive synthetic biology demonstration that took it all the way to cancer cell lines. However, given the sole demonstrated output in the form of fluorescent proteins, the authors' claims related to therapeutic implications require additional empirical evidence or, otherwise, expository revision.

Thank you very much for the thoughtful evaluation, precise critique, and constructive suggestions.

As correctly noted, our study initially focused on developing and optimizing input sensors and processing units for synthetic gene circuits targeting mutated RAS. To address the concern regarding therapeutic relevance, we have now incorporated functional validation using a clinically relevant output protein: herpes simplex virus thymidine kinase (HSV-TK), which converts ganciclovir into a cytotoxic compound. We replaced the mCerulean reporter with HSV-TK and tested the resulting RAS-targeting circuits in both RAS-mutant and wild-type cancer cell lines. The results, now presented in a new chapter (Figure 8 and Supplementary Fig. 14), demonstrate robust killing of RAS-mutant cells and support the potential therapeutic utility of these circuits.

Major comments:"These therapies are limited to cancers with KRASG12C mutations" is technically accurate. However, in this fast-moving field, there are examples such as MRTX1133 which holds the promise to target the very G12D mutation that is the focus of this paper. There are broader efforts too. It would help the readers better appreciate the background if the authors could update the intro to reflect the most recent landscape of RAS-targeting drugs.

Thank you for this helpful suggestion. We have updated the introduction to reflect the rapidly evolving landscape of RAS-targeting therapies, including the development of inhibitors for nonG12C mutations such as KRASG12D (e.g., MRTX1133). Given the pace and breadth of these advances, we also refer readers to a recent comprehensive review that provides an in-depth overview of current RAS-targeting strategies.

Only KRASG12D was used as a model in the design and optimization work of the genetic circuits. Other mutations should be quite experimentally feasible and comparisons of the circuits' performances across different KRAS mutations would allow for stronger claims on the circuits' generalizability. Particularly, the cancer cell line used for circuit validation harbored a KRASG13D mutation. While the data presented do indeed support the circuit's "generalizability," the model systems would not have been consistent in the current set of data presented.

To further support the generalizability of our RAS sensor, we titrated plasmid doses for a panel of oncogenic RAS variants, including multiple KRAS mutants as well as HRAS^G12D</sup and NRASG12D</sup. Across all tested variants, we observed concentration-dependent activation of the RAS sensor. At 1.67 ng/well, the sensor output for all oncogenic RAS variants was at least as high as that for KRASG12D, suggesting that the behavior observed in our initial design and optimization is representative of a broader set of RAS mutations.^

We also noted that high overexpression of wildtype HRAS and NRAS can lead to substantial activation of the sensor, exceeding that observed with wildtype KRAS. This underscores the importance of considering all RAS isoforms when assessing circuit specificity and avoiding potential off-target activation in healthy cells.

In Figure 2a, the text claims that "inactivation of endogenous RAS with NF1 resulted in a lower YFP/RBDCRD-NarX expression," but Figure 2a does not show a statistically significant reduction in expression of SYFP (measured by "membrane-to-total signal ratio [RU]).

Thank you for pointing this out. We repeated the experiment to reassess the effect of NF1 on RBDCRD-NarX-SYFP2 expression and were able to confirm statistical significance. Accordingly, we have replaced Figure 2a with updated data. To facilitate better visual comparison across conditions, we also standardized the y-axis range across all relevant flow cytometry plots.

The therapeutic index of the authors' systems would be better characterized by a functional payload, other than florescent proteins, that for example induce cell death, immune responses, etc.

Thank you for this insightful comment. We agree that fluorescent reporters are limited to approximating expression levels, and that a functional output protein is more appropriate for assessing therapeutic potential. To address this, we replaced mCerulean with the therapeutic suicide-gene, HSV-TK, and tested the circuits in RAS-mutant and wild-type cancer cell lines. These experiments demonstrate that our circuits can express functional proteins and induce cell death in two RAS-mutant cell lines while showing low toxicity in a RAS wild type cell line (new chapter including Fig. 8 and Supplementary Fig.14).

Comparing confluence of cells transfected with the RAS-targeting circuits to cells transfected with non-toxic GFP-output negative control or the constitutively expressed EF1αHSV-TK positive control allowed us to estimate the killing-strength of the circuits in each cell line. In RAS-mutant HCT-116 the confluence curves were similar to the positive control, indicating effective killing (Fig. 8b). At lower DNA dose in HCT-116, or in SW620 with lower transfection efficiency, the killing of transfected RAS-driven cancer cells was less pronounced, falling approximately midway between the controls (Fig. 8g&j). In the RAS wild type cell line, Igrov-1, the RAS circuits showed continued growth similar to the non-toxic negative control (Fig. 8d), suggesting low toxicity.

While this may indicate low circuit activation in Igrov-1, an alternative explanation for the low toxicity could also be insufficient transfection efficiency. Testing in SW620 –which had similar transfection efficiency as Igrov-1 (Supplementary Fig. 14a)– showed that this moderate transfection efficiency was sufficient for RAS-circuit-dependent killing (Fig. 8d & 8g), supporting the notion of low activation in Igrov-1 and selective cytotoxicity in RAS-driven cancer cells.

Nonetheless, it is important to note that comparisons between the cell lines need to be interpreted cautiously because of inter-cell line differences in transfection, growth, and HSV-TK/ganciclovir (GCV)-sensitivity (Supplementary Fig. 14) and further validation will be essential.

A conclusive assessment will require more efficient delivery strategies, such as viral vectors (as discussed above). Efficient delivery would allow to investigate selectivity in a more realistic setting with patient-derived RAS-mutant cancer and healthy cells as well as testing in an vivo model. While beyond the scope of the current study, we view it as a critical direction for future work and have therefore added a paragraph about this to our discussion.

Regarding data presented in "Mechanism of action" (Figure 2), the observations are interesting and consistent across different fluorescent reporters. However, with regard to interpretations of the underlying molecular mechanisms, it is not clear whether the different output levels in 2b, 2c, and 2d are due to the pathway as described by the authors or simply from varied expression levels of RBDCRD-NarX itself (2a) that is nonlinearly amplified by the rest of the circuit. From a practical standpoint, this caveat is not critical with respect to the signal-to-noise ratios in later parts of the paper. From a mechanistic interpretation standpoint, claims made forth in this section are not clearly substantiated. Some additional controls would be nice. For example, if the authors express NarXs that constitutively dimerize on the membrane, what would the RasG12Dresponsiveness look like? Does RasG12D alter the input-output curve of NarL-RE? How would Figure 4f compare to a NaxR constitutively dimerized control that only relies on transcriptional amplification of the Ras-dependent promoters?

This is a great point. We agree that the observed differences in output levels (Fig. 2) could arise from non-linear amplification due to increased expression of RBDCRD-NarX, rather than RAS binding or dimerization alone. To further investigate this possibility, we performed titrations of KRAS^G12D^ in combination with the functional RAS sensor and a series of constitutively active and inactive control constructs (Supplementary Fig. 4).

Inactive controls lacking NarX dimerization showed only a modest increase in output expression, similar to direct mCerulean expression under the EF1α promoter. Transfection of the output plasmid alone, with NarL, or with NarL and non-RAS-binding RBD^R89L^ CRD^C168S^ -NarX, resulted in minimal RAS-dependent increases (Supplementary Fig. 4a). Importantly, after normalization using the EF1α-driven mCherry transfection control, these effects were fully or even slightly over-compensated (Supplementary Fig. 4b), showing that we don’t include the effect of EF1α-dependent increased leakiness in the data presented throughout the manuscript, but also that –due to the normalization– we potentially underestimate the dynamic range of the RAS-targeting circuits.

In contrast, constitutively dimerizing NarX controls (both membrane-bound and cytosolic dimerized via the FKBP–FRB system) exhibited a more pronounced RAS-dependent increase in output –even after normalization– confirming the presence of non-linear amplification (up to 3–4fold). However, this effect was still lower than that achieved with the functional RAS-binding sensor (8-fold at 1.67 ng/well KRAS^G12D^; 14-fold at 5–15 ng/well), indicating that the increase in expression of the sensor parts is not the full explanation of the effect we see. Instead, RAS binding and dimerization further amplify the response and are necessary for full activation (Supplementary Fig. 4b).

We also addressed the reviewer’s suggestion by testing the MAPK response elements used in Fig. 4f with constitutively dimerizing NarX. These controls generally showed lower fold changes between KRAS^G12D^; and KRAS^WT^ than the corresponding RAS-binding circuits (Supplementary Fig. 7), with one exception: the combination of SRE_NarX and PY2_NarL-VP48.

Together, these data show that non-linear amplification via increased expression and dimerization contributes to output activation. However, RAS binding and induced dimerization of the NarX sensor are required for full functionality and enhanced signal strength. This underscores that integrating the MAPK response elements with the binding-based RAS sensor into RAS-targeting circuits generally improves the distinction between cells with KRAS^G12D^; and KRAS^WT^ and that it was the combination that allowed to reach maximal fold changes.

It's also possible that these Ras could affect protein production at the post-transcriptional or even post-translational levels, which were not adequately considered.

Thank you for this comment. We now mention in the manuscript the potential mechanisms by which (over-)activated RAS or MAPK signaling can increase protein synthesis. We cite relevant reports of the mechanisms we found, including upregulation of translational initiation and machinery[10] and ribosomal biogenesis[11].

The text claims that "in contrast to what we saw in HEK293 overexpressing RAS (Figure 5d), the "AND-gate" RAS-targeting circuits do not generate higher output than the EF1a-driven, bindingtriggered RAS sensor in HCT-116. Instead, the improved dynamic range results from decreased leakiness in HCT- 116k.o." Comparing the experiment from Figure 5d, which looks at activation in KRASG12D and KRASWT, to the experiments in Figure 6b-d, which looks at activation in HCT-116WT and HCT-116KO is misleading. In Fig 5d., cells are transfected with KRASG12D and KRASWT to emulate high levels of mutant RAS and high levels of wild-type RAS. In Figures 6b-d, HCT-116WT has endogenous levels of mutant RAS, while the KCT-116KO is a knock-out cell line, and does not have mutant or WT RAS. Therefore, the improved dynamic range or "decreased leakiness in HCT-116KO" in comparison to Figure 5d. is more comparable to the NF1 condition from Figure 2, which deactivates endogenous RAS. While this may not be feasible, the most accurate comparison would have been an HCT-116KO line with KRASWT stably integrated.

Thank you for this input. We understand that comparing the results from HEK293 cells transfected with KRAS^G12D^; or KRAS^WT^ (Fig. 5d) to those from HCT-116^WT^ and HCT-116^k.o^. cells (Fig. 6b–d) may be misleading if interpreted as a direct comparison of RAS signaling levels. Our intent was not to compare HEK293 with KRAS^WT^ directly to HCT-116^k.o^.., but rather to contrast the behavior of the EF1α-driven RAS sensor and the MAPK-responsive RAS-targeting circuits within each cell line context.

Specifically, we observed that in HEK293 cells expressing KRAS^G12D^, the MAPK-based RAS-targeting circuits produced higher output than the EF1α-expressed RAS sensor. In contrast, in HCT-116^WT^ cells, the EF1α-expressed RAS sensor resulted in higher output levels than the RAS-targeting circuits. Despite this, the MAPK-driven circuits showed an improved dynamic range compared to the EF1α-expressed RAS sensor in HCT-116, due to the reduced background expression in the HCT-116^k.o^.. cells. We have revised the manuscript text to clarify this distinction.

We agree that an HCT-116^k.o^ cell line with stable integration of KRAS^WT^ would provide a more direct comparison. Nonetheless, HCT-116^k.o^.. cells still express endogenous NRAS and HRAS, both of which are capable of activating the RAS sensor (as shown in Fig. 1g). Therefore, we believe that HCT-116^k.o^. cells are more comparable to HEK293 with KRAS^WT^ than to the NF1 condition in Fig. 2, in which all endogenous RAS isoforms are inactivated.

We couldn't locate the citation or discussion of Figure 4d in the text. Conversely, based on the text description, Figure 6g would contain exciting results. But we couldn't find Figure 6g anywhere ... unless it was a typo and the authors meant Figure 6f, in which case the cool results in Figure S8 could use more elaboration in the main text.

Thank you for this helpful observation. The figure references were indeed incorrect due to a typo. The results discussed in the text refer to Figure 6f (not 6g), which is now Figure 7a in the revised version. To further highlight these findings, we have added a new Figure 7b that better illustrates how different MAPK response elements enabled us to identify, for each RAS-mutant cell line, a RAS-targeting circuit that showed stronger activation than in all RAS wild-type lines. We have also expanded the corresponding section in the main text to elaborate on these results and their significance.

**Reviewer #3 (Public review):**
Summary:Mutations that result in consistent RAS activation constitute a major driver of cancer. Therefore, RAS is a favorable target for cancer therapy. However, since normal RAS activity is essential for the function of normal cells, a mechanism that differentiates aberrant RAS activity from normal one is required to avoid severe adverse effects. To this end, the authors designed and optimized a synthetic gene circuit that is induced by active RAS-GTP. The circuit components, such as RAS-GTP sensors, dimerization domains, and linkers. To enhance the circuit selectivity and dynamic range, the authors designed a synthetic promoter comprised of MAPK-responsive elements to regulate the expression of the RAS sensors, thus generating a feed-forward loop regulating the circuit components. Circuit outputs with respect to circuit design modification were characterized in standard model cell lines using basal RAS activity, active RAS mutants, and RAS inactivation.This approach is interesting. The design is novel and could be implemented for other RASmediated applications. The data support the claims, and while this circuit may require further optimization for clinical application, it is an interesting proof of concept for targeting aberrant RAS activity.Strengths:Novel circuit design, through optimization and characterization of the circuit components, solid data.Weaknesses:This manuscript could significantly benefit from testing the circuit performance in more realistic cell lines, such as patient-derived cells driven by RAS mutations, as well as in corresponding non-cancer cell lines with normal RAS activity. Furthermore, testing with therapeutic output proteins in vitro, and especially in vivo, would significantly strengthen the findings and claims.

Thank you very much for the thoughtful and supportive comments. We fully agree with the reviewer’s suggestions for improving the translational potential of the RAS-targeting circuits.

As a first step toward therapeutic relevance, we replaced the fluorescent reporter with HSV-TK, a clinically validated suicide gene, and demonstrated killing in RAS-mutant cancer cell lines. This is described above and in the new section of the manuscript (Figure 8).

We also agree that testing in patient-derived cancer cells and especially healthy cells with wild-type RAS activity will be essential. However, testing in primary or patient-derived cells presents delivery challenges: transient transfection of our current four-plasmid system is unlikely to achieve sufficient expression. As discussed in our response to Reviewer #1, development of a more efficient delivery strategy –such as viral vector-based delivery– is a necessary next step.

Once a delivery system is established, identifying relevant off-target tissues throughout the body with high physiological RAS signaling will be key to assessing selectivity. While comparative data on RAS activation across healthy tissues are scarce[12,13], recent atlases of transcription factor activity[14,15] provide insights to identify off-target cells with high activation of RAS-dependent transcription factors and may even approximate RAS activity across healthy tissue. Alternatively, our single-input sensors for RAS and MAPK pathway activity could be used in vivo to identify off-target cells based on endogenous activity.

Once relevant target and off-target cells have been identified, patient-derived cancer and healthy cells can help select and adapt cancer-specific RAS-targeting circuits and nominate therapeutic candidates for further safety and efficacy assessment[6,8].

**Reviewer #1 (Recommendations for the authors):**
For the most part, the data in this study are very convincing and very well presented. The cartoons make it easier to understand the complex experimental setups.(1) Did the authors use wild-type Sos-1 or a constitutively active membrane-bound catalytic domain in their studies? How is SOS-1 activated when in case Sos-1 wild-type was used?

Thank you for this feedback. We used the constitutively active catalytic domain of Sos-1 (AA5641049; PDB ID 2II0).

(2) Figure 1f: In case of KRAS-G12D, it looks like the output expression does not really correlate with the RAS-GTP level. Can the authors give an explanation?

Thank you for this interesting question. We believe the observed discrepancy arises primarily from differences in the sensitivity and readout dynamics of the two assays. The RAS-GTP pulldown ELISA appears insufficiently sensitive to detect small changes in RAS-GTP levels at lower KRAS^G12D^ plasmid doses (0.19, 0.56, or 1.67 ng). Only at 5 ng and 15 ng do we observe clear increases in RAS-GTP signal (25% and 700%, respectively). In contrast, the RAS sensor shows strong activation already in the 0.56–5 ng range but begins to saturate at higher doses (see Figure 1f and Figure 1e).

Beyond the differing technical sensitivities of the ELISA (plate reader) and flow cytometry, an important conceptual distinction may further explain this behavior: the RAS sensor likely integrates RAS signaling over time. Once NarX binds RAS-GTP and dimerizes, it activates NarL, triggering mCerulean expression. If the rate of mCerulean production exceeds its degradation, signal accumulates throughout the assay duration. Thus, the flow cytometry readout reflects time-integrated signaling, allowing small differences in RAS-GTP to be amplified into measurable differences in output—especially at low input levels. This may explain why flow cytometry detects circuit activation earlier and more steeply than the pulldown assay, which provides a snapshot of RAS-GTP abundance at a single time point and saturates less readily at high input levels.

Together, these factors likely explain the observed differences in signal dynamics: the RAS sensor exhibits steep activation followed by saturation at high plasmid doses (flow cytometry), while the ELISA shows limited sensitivity at low doses but a broader linear range at higher doses.

(3) Figure 2b: It appears that even in the case of KRAS-G12D and Sos-1, only a few cells are positive. Does this result depend on low cell density, low transfection efficiency, or a wide range of the expression level? As a control, nuclear staining could be shown.

Thank you for this question. In the experiment shown in Figure 2b, our goal was to assess the membrane localization of the RBD^CRD-NarX-SYFP2 construct, which serves as a proxy for RAS-bound sensor. To enable accurate computational segmentation and separation of membrane signal from adjacent cells, we intentionally reseeded cells at low density in glassbottom plates for confocal imaging.

The observed variability in signal likely reflects a combination of transient transfection and heterogeneous expression levels. While the overall transfection efficiency was approximately 70%, expression varied between individual cells. To account for this, we analyzed the membrane-to-total signal ratio per cell, which internally normalizes the membrane signal to the total cellular expression of SYFP2 and controls for differences in transfection efficiency.

In response to the reviewer’s suggestion, we have updated the figure to include nuclear staining to aid interpretation. We would like to emphasize, however, that the images are intended to illustrate subcellular localization per cell, not expression frequency or intensity across the population.

Minor points(1) Figure 1b: "The third plasmid expresses NarL, .." should be changed to "The third plasmid expresses NarL-VP48, .."

Done

(2) Figure 1c, right part: The orange arrow should be labeled NarX-H399Q (not N509A).

Done

(3) Supplementary Table 6 and 7: [cells/wells] - should probably be [cells 10*3/well].

Thank you for these points, we updated the manuscript accordingly

**Reviewer #2 (Recommendations for the authors):**
Minor comments:(1) N509A seems mislabeled in Figure 1b.(2) It would help the readers if the authors could elaborate a bit on what is known about the RBD and CRD mutations used here.

Thank you for the input, we added a paragraph in the paper to expand on the effect of these commonly used mutations.

(3) The KRASWT&Sos1 condition is not explained within the text for Figure 1f, which is the first figure with the KRASWT&Sos1 condition, but rather later on for Figure 2a. Adding a description of this condition to the discussion of Figure 1f would add clarity to this figure.

Thank you, we corrected this.

(4) Citing AlphaFold2 structural predictions as having "revealed that longer linkers between the sensor's RBDCRD and NarX-derived domains could bring the NarX domains into closer proximity" is probably an overstatement. AlphaFold2 generally has low confidence in the placement of long flexible linkers, and the longer linkers in the illustration could facilitate NarX and NarL being even farther apart than they are in the original design.

Thank you for this input. We agree that AlphaFold2 predictions generally have low confidence in the placement of long, flexible linkers, and we did not intend to imply that the structural models were predictive of actual linker conformations. Rather, the models were used heuristically to generate the hypothesis that longer linkers might facilitate better positioning of the NarX domains for dimerization.

As described in the Methods, we manually rotated the flexible linker regions to explore plausible conformations. These exploratory models showed that with a short (1x GGGGS) linker, it was more challenging to bring the NarX domains into close proximity, whereas longer linkers allowed greater positional flexibility. This modeling exercise provided a structural rationale for experimentally testing longer linkers. We have revised the manuscript text to clarify that the structural predictions were used to motivate linker design –not to validate or predict structural outcomes.

(5) Figure 3b shows that the fold change (KRASG12D/KRASWT) is higher at shorter linker lengths and lower at longer linker lengths, and that the output expression of mCerulean is lower at shorter linker lengths and higher at longer linker lengths. Having a bar plot with the output expression mCerulean levels comparing KRASG12D and KRASWT next to each other would be a significantly more informative representation of this data. In particular, the readers might be interested in understanding the effect of linker length on off-target activation from the sensor, which is not clear from this figure.

Thank you for the suggestion. We adapted Figure 3b to better present this.

(6) While it is implied that the sentence "Among the tested binding domains, the Ras association domain (RA) of the natural RAS effector Rassf5, the RAS association domain 2 (RA2) of the phospholipase C epsilon (PLCe)33, and the synthetic RAS binder K5534 showed a slightly higher or similar dynamic range." is comparing these RAS binding domains to RBDCRD, for clarity it should be noted what the point of reference is for this "slightly higher or similar dynamic range."(7) Claims are made throughout the text that require supporting data, and thus require a reference to a figure, but there are a few instances where the reference is several sentences after the discussion of data and findings begins. For example, the discussion of Figure 3c begins with the claim "Among the tested binding domains, the Ras association domain (RA) of the natural RAS effector Rassf5, the RAS association domain 2 (RA2) of the phospholipase C epsilon (PLCe)33, and the synthetic RAS binder K5534 showed a slightly higher or similar dynamic range," but there is no reference to the data or figure being discussed until the end of the discussion of Figure 3c. This formatting is also present in Figure 3d and Figure 6f.

Thank you for mentioning these imprecisions and inconsistencies, we addressed them in the manuscript.

(8) In Figures 5d and 5e, the formatting of underscores and dashes is occasionally inconsistent within the text. (ex. "PY2_NarX_FLT or PY2_NarL-FLT" on page 13.).

Thank you for this precise observation. The formatting differences were intentional and reflect distinct design principles. Specifically:

An underscore (e.g., PY2_NarX_FLT) denotes that two separate proteins are expressed –here, PY2-driven RBDCRD-NarX and EF1α-driven NarL-F.L.T.

A dash (e.g., PY2_NarL-F.L.T.) indicates a fusion protein –i.e., PY2-driven NarL-F.L.T. combined with EF1α-driven RBDCRD-NarX.

This notation is used to distinguish expression sources and fusion constructs while avoiding redundancy with the base circuit (EF1α_NarX + EF1α_NarL-VP48). We hope the included schematic diagrams in each relevant figure helps the reader interpret these combinations.

(9) The text claims that "loss-of-function mutations in RBDCRD decreased activation. However, the dynamic range was only 3-fold" and attributes this claim to Figure 6a. For a claim about specific fold-change activation, one would expect a corresponding figure with quantitative measurements of this fluorescence to be referenced.

Thank you for this remark. We made a supplementary figure (Supplementary Fig. 11) to show the quantitative measurement of the 3-fold dynamic range between HCT-116^WT^ and HCT-116^k.o^. when using the EF1a-expressed RAS sensor with NarL-VP48.

(10) The claim of this Figure 2d is that the effect of RAS-GTP levels on mCerulean output is amplified in comparison to Figures 2a, 2b, and 3c, representing expression, RAS binding, and dimerization respectively. While visually this might be true from the figure, the readers might be confused by the lack of significance between the control and the NF1 condition, alongside the variation between the triplicates. Could this experiment be repeated to gain clearer data and to support their claim more effectively?

Thank you for this important observation. To address the concern regarding variability and statistical significance in Figure 2d, we repeated the experiment using 24-well plates to increase the number of cells analyzed per condition. This improved the consistency of the data and allowed us to reduce variability across replicates. As a result, we now observe a statistically significant difference between the control and the NF1 condition. The updated results are shown in the revised Figure 2.

(11) The readers might be less familiar with the concept of "composability" than "modularity" and it would be good to explain it if the authors did intend to use the former.

Thank you for this comment. We changed it to modularity to avoid confusion.

References

(1) Shahryari, A., Burtscher, I., Nazari, Z. & Lickert, H. Engineering Gene Therapy: Advances and Barriers. Advanced Therapeutics vol. 4 Preprint at https://doi.org/10.1002/adtp.202100040 (2021).

(2) Mcclements, M. E. & Maclaren, R. E. Adeno-Associated Virus (AAV) Dual Vector Strategies for Gene Therapy Encoding Large Transgenes. YALE JOURNAL OF BIOLOGY AND MEDICINE vol. 90 (2017).

(3) Wagner, H. J., Weber, W. & Fussenegger, M. Synthetic Biology: Emerging Concepts to Design and Advance Adeno-Associated Viral Vectors for Gene Therapy. Advanced Science vol. 8 Preprint at https://doi.org/10.1002/advs.202004018 (2021).

(4) Doshi, J., Willis, K., Madurga, A., Stelzer, C. & Benenson, Y. Multiple Alternative Promoters and Alternative Splicing Enable Universal Transcription-Based Logic Computation in Mammalian Cells. Cell Rep **33**, 108437 (2020).

(5) Wu, Z., Yang, H. & Colosi, P. Effect of genome size on AAV vector packaging. Molecular Therapy **18**, 80–86 (2010).

(6) Dastor, M. et al. A Workflow for in Vivo Evaluation of Candidate Inputs and Outputs for Cell Classifier Gene Circuits. ACS Synth Biol **7**, 474–489 (2018).

(7) Preuß, E. et al. TK.007: A novel, codon-optimized HSVtk(A168H) mutant for suicide gene therapy. Hum Gene Ther **21**, 929–941 (2010).

(8) Angelici, B., Shen, L., Schreiber, J., Abraham, A. & Benenson, Y. An AAV gene therapy computes over multiple cellular inputs to enable precise targeting of multifocal hepatocellular carcinoma in mice. Sci Transl Med **13**, (2021).

(9) Mesnil, M. & Yamasaki, H. Bystander Effect in Herpes Simplex Virus-Thymidine Kinase/Ganciclovir Cancer Gene Therapy: Role of Gap-Junctional Intercellular Communication 1. CANCER RESEARCH vol. 60 http://aacrjournals.org/cancerres/articlepdf/60/15/3989/2478218/ch150003989.pdf (2000).

(10) Proud, C. G. Ras, PI3-kinase and mTOR signaling in cardiac hypertrophy. Cardiovascular Research vol. 63 403–413 Preprint at https://doi.org/10.1016/j.cardiores.2004.02.003 (2004).

(11) Azman, M. S. et al. An ERK1/2driven RNAbinding switch in nucleolin drives ribosome biogenesis and pancreatic tumorigenesis downstream of RAS oncogene. EMBO J **42**, (2023).

(12) von Lintig, F. C. et al. Ras activation in normal white blood cells and childhood acute lymphoblastic leukemia. Clin Cancer Res **6**, 1804–10 (2000).

(13) Guha, A., Feldkamp, M. M., Lau, N., Boss, G. & Pawson, A. Proliferation of human malignant astrocytomas is dependent on Ras activation. Oncogene **15**, 2755–2765 (1997).

(14) Pan, L. et al. HTCA: a database with an in-depth characterization of the single-cell human transcriptome. Nucleic Acids Res **51**, D1019–D1028 (2023).

(15) Pan, L. et al. Single Cell Atlas: a single-cell multi-omics human cell encyclopedia. Genome Biol **25**, (2024).